# Growth inhibitory factor/metallothionein-3 is a sulfane sulfur-binding protein

Yasuhiro Shinkai[1,2]*, Yunjie Ding[3], Toru Matsui[4], George Devitt[5], Masahiro Akiyama[2], Tang-Long Shen[6], Motohiro Nishida[7,8], Tomoaki Ida[9], Takaaki Akaike[9], Sumeet Mahajan[10], Jon M Fukuto[11], Yasuteru Shigeta[12], Yoshito Kumagai[2,7]

[1]Environmental Biology Laboratory, School of Life Sciences, Tokyo University of Pharmacy and Life Sciences, Hachioji, Japan; [2]Environmental Biology Laboratory, Faculty of Medicine, University of Tsukuba, Tsukuba, Japan; [3]Doctoral Program in Biomedical Sciences, Graduate School of Comprehensive Human Sciences, University of Tsukuba, Tsukuba, Japan; [4]Department of Chemistry, Graduate School of Pure and Applied Sciences, University of Tsukuba, Tsukuba, Japan; [5]Centre for Biological Sciences, University of Southampton, Highfield Campus, Southampton, United Kingdom; [6]Department of Plant Pathology and Microbiology, National Taiwan University, Taipei, Taiwan; [7]Graduate School of Pharmaceutical Sciences, Kyushu University, Fukuoka, Japan; [8]National Institute for Physiological Sciences & Exploratory Research Center on Life and Living Systems, National Institutes of Natural Sciences, Okazaki, Japan; [9]Department of Environmental Medicine and Molecular Toxicology, Tohoku University Graduate School of Medicine, Sendai, Japan; [10]School of Chemistry and the Institute for Life Sciences, University of Southampton, Highfield Campus, Southampton, United Kingdom; [11]Department of Chemistry, Johns Hopkins University, Baltimore, United States; [12]Center for Computational Sciences, University of Tsukuba, Tsukuba, Japan

*For correspondence:
yshinkai@toyaku.ac.jp

Competing interest: The authors declare that no competing interests exist.

## eLife Assessment

This **valuable** work provides **solid** evidence that a neuronal metallothionein, GIF/MT-3, incorporates metal-persulfide clusters. A variety of well-designed assays support the authors' hypothesis, revealing that sulfane sulfur is released from MT-3. However, the sufane sulfur content in the canonical induced MT-1 and MT-2 has not been demonstrated. Thus, the biological role of the persulfidated form is not yet clearly defined. There are caveats to the findings that limit the study, but the work will nevertheless prompt major follow-up work.

**Abstract** Cysteine-bound sulfane sulfur atoms in proteins have received much attention as key factors in cellular redox homeostasis. However, the role of sulfane sulfur in zinc regulation has been underinvestigated. In this study, we identified growth inhibitory factor (GIF)/metallothionein-3 (MT-3) as a sulfane sulfur-binding protein from mouse brain. We also report here that cysteine-bound sulfane sulfur atoms serve as ligands to hold and release zinc ions in GIF/MT-3 with an unexpected C–S–S–Zn structure. Oxidation of such a zinc/persulfide cluster in $Zn_7$GIF/MT-3 results in the release of zinc ions, and intramolecular tetrasulfide bridges in apo-GIF/MT-3 efficiently undergo S–S bond cleavage by thioredoxin to regenerate $Zn_7$GIF/MT-3. Three-dimensional molecular modeling

confirmed the critical role of the persulfide group in the thermostability and Zn-binding affinity of GIF/MT-3. The present discovery raises the fascinating possibility that the function of other Zn-binding proteins is controlled by sulfane sulfur.

## Introduction

Sulfane sulfur is a chemical state of the sulfur atom with six valence electrons that are covalently bound to sulfur atoms (*Toohey, 2011*; *Iciek et al., 2019*). Growing evidence supports the widespread existence of hydropersulfidated and polysulfidated proteins in all cell types, referred to as sulfane sulfur-binding proteins (SSBPs) (*Shinkai and Kumagai, 2019*). Protein sulfuration occurs via post-translational and co-translational pathways. Rhodanese is known to catalyze the production of sulfane sulfur and attach it to the thiol group of the protein itself using thiosulfate as a substrate (*Kruithof et al., 2020*). 3-Mercaptopyruvate sulfurtransferase is a rhodanese-like enzyme that uses 3-mercaptopyruvate as the preferred sulfur donor (*Pedre and Dick, 2021*). Cystathionine β-synthase and cystathionine γ-lyase use cystine as a substrate and catalyze the production of sulfane sulfur-containing cysteine hydropersulfide (CysSSH) (*Ida et al., 2014*), whose terminal sulfane sulfur can be reversibly transferred to other thiols such as glutathione (GSH) or protein-SH to form GSH hydropersulfide (GSSH) or protein hydropersulfides (protein-SSH), respectively (*Fukuto et al., 2018*). Cysteinyl-tRNA synthetase 2 catalyzes the production of CysSSH from CysSH (*Akaike et al., 2017*), thereby producing CysSSH-integrated nascent proteins. The biological function of sulfane sulfur has received considerable attention in redox biology because of its antioxidant/anti-electrophilic capacity. However, the role of sulfane sulfur in proteins is not fully understood; therefore, further mechanistic investigation is required. Although several SSBPs have been identified using various methods (*Ida et al., 2014*; *Abiko et al., 2015*), in the present study, we used β-(4-hydroxyphenyl)ethyl iodoacetamide (HPE-IAM) (*Akaike et al., 2017*) to derivatize the sulfane sulfur because we herein found that HPE-IAM has the ability to extract sulfane sulfur atoms from SSBPs to form bis-S-β-(4-hydroxyphenyl)ethyl acetamide (bis-S-HPE-AM) adduct at a certain condition, thereby enabling quantitative analysis using LC–MS/MS with a stable isotope-labeled standard, bis-S$^{34}$-HPE-AM.

In biological systems, cysteine-rich proteins can act as 'redox switches', which sense accumulated oxidative stressors and free zinc ions, store excess metals, control the activity of metalloproteins, and serve as triggers for the activation of cellular redox signaling cascades (*Giles et al., 2003*). Metallothionein (MT), discovered in 1957 (*Margoshes and Vallee, 1957*), is an important cysteine-rich metal-binding protein involved in three major biological processes: homeostasis of essential metals, detoxification of toxic metals, and protection from oxidative stress (*Kang, 2006*; *Maret, 2008*). It is recognized that metal binding to MT is thermodynamically stable, but oxidation of the thiolate cluster readily leads to metal release and formation of intramolecular MT–disulfide linkages. Zinc ions released from zinc/thiolate clusters of MT are suggested to function as signaling molecules for cellular redox homeostasis (*Krezel et al., 2007*). Simultaneously, reduction of MT–disulfide by cellular reducing agents can occur in a process called the 'MT redox cycle' (*Kang, 2006*). However, the biochemical features of MT related to these functions have not been fully characterized. In addition, although the gas chromatography–flame photometric detector technique showed that MT isoforms contain sulfide ions (*Capdevila et al., 2005*; *Tío et al., 2006*), it remains unclear if these sulfides are indeed sulfane sulfur atoms that act as essential factors in controlling protein redox states, thereby regulating cellular zinc homeostasis. Because of its constitutive expression, this study focused on MT-3, which was originally identified as a growth inhibitory factor (GIF) in the human brain (*Uchida et al., 1991*). This study aimed to clarify the existence and content of sulfane sulfur in GIF/MT-3, the redox regulation of sulfane sulfur in holo- and apo-GIF/MT-3 in association with zinc release, and the effect of sulfane sulfur on the thermostability and metal-binding affinity of GIF/MT-3. We found that sulfane sulfur atoms provide a redox-dependent switching mechanism for zinc/persulfide cluster formation in GIF/MT-3.

## Results

### Existence of persulfide and polysulfide groups in apo-GIF/MT-3

It is well recognized that GIF/MT-3 is able to bind seven zinc ions. To determine if GIF/MT-3 is an SSBP, we used an *Escherichia coli* expression system to prepare recombinant human Zn$_7$GIF/MT-3 protein,

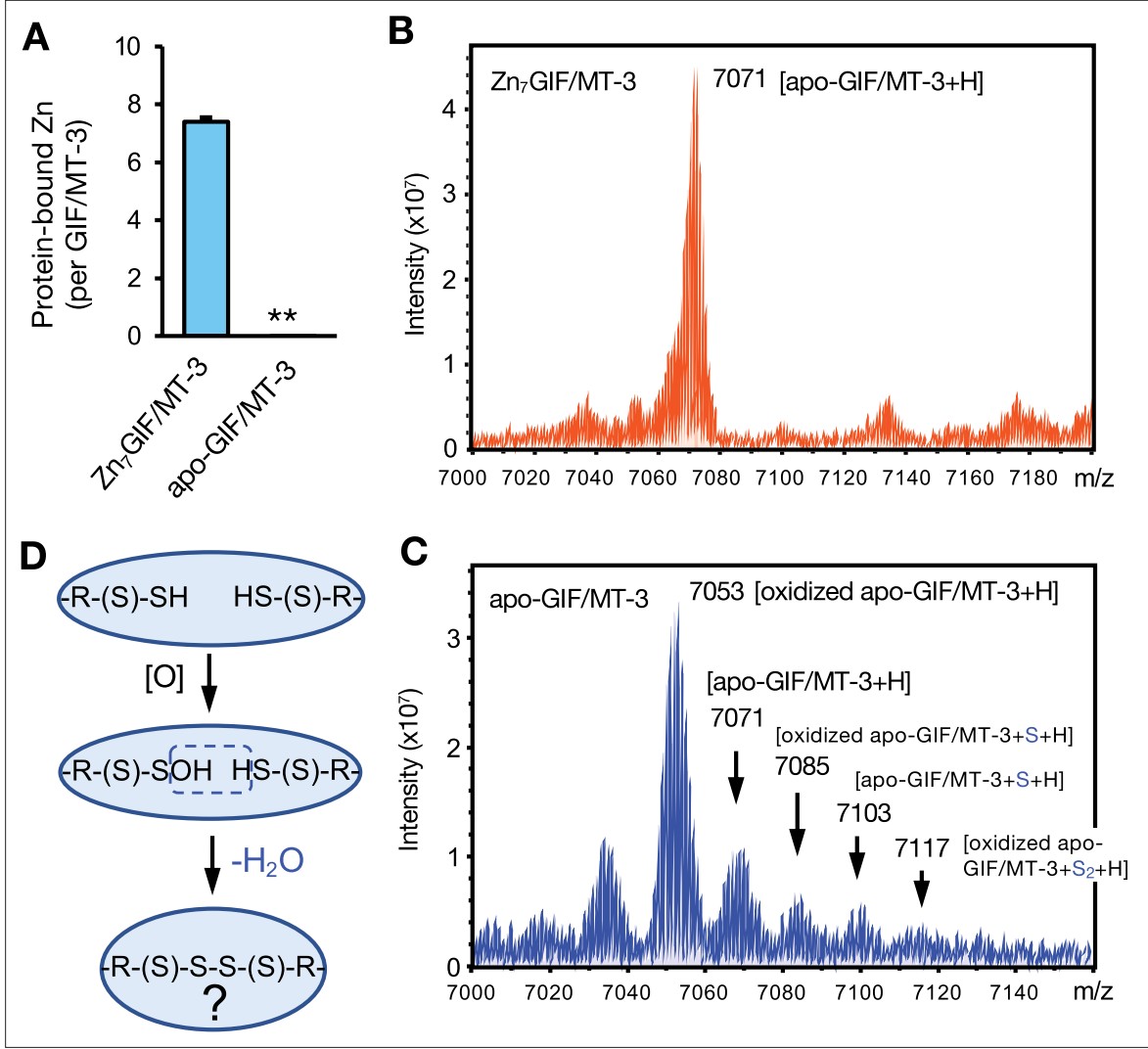

**Figure 1.** Detection of sulfane sulfur in GIF/MT-3 by MALDI–TOF/MS. (**A**) Preparation of recombinant $Zn_7$GIF/MT-3 and oxidized apo-GIF/MT-3 proteins. Recombinant human $Zn_7$GIF/MT-3 (10 μM) was incubated in HCl (0.1 N) at 37°C for 30 min and then replaced with 20 mM Tris–HCl (pH 7.5) buffer and incubated for 36 hr at 37°C. After removal of low-molecular-weight molecules using 3 kDa centrifugal filtration, GIF/MT-3-bound zinc content was measured using ICP-MS. Each value represents the mean ± SD of three independent experiments. (**B**) FT–ICR–MALDI–TOF/MS spectrum (positive-ion mode) of $Zn_7$GIF/MT-3. (**C**) FT–ICR–MALDI–TOF/MS spectrum (positive-ion mode) of oxidized apo-GIF/MT-3. (**D**) Putative oxidation reaction scheme in apo-GIF/MT-3 protein. FT-ICR, Fourier transform ion cyclotron resonance.

The online version of this article includes the following source data for figure 1:

**Source data 1.** Source data for panel A: Zinc content measured in GIF/MT-3 samples.

from which apo-GIF/MT-3 was subsequently prepared (***Figure 1A***). To detect sulfur modification in GIF/MT-3, we attempted to first measure the molecular weight of the whole protein with and without bound zinc. Using Fourier transform ion cyclotron resonance (FT–ICR)–MALDI–TOF/MS, $Zn_7$GIF/MT-3 was mainly detected at $m/z$=7071 (***Figure 1B***), which corresponded to the mass of zinc-free apo-GIF/MT-3 and indicated that zinc dissociates from protein in the acidic conditions used for MALDI sample preparation. However, apo-GIF/MT-3 showed several peaks at $m/z$=7071, 7085, 7103, and 7117, with a main peak at $m/z$=7053 (***Figure 1C***) that corresponded to oxidized GIF/MT-3 with nine intramolecular cysteine disulfide bonds, which presumably release the molecular weight equivalent of nine molecules of $H_2$ (***Figure 1D***). Thus, the peaks at $m/z$=7085 (oxidized GIF/MT-3 plus one sulfur atom), 7103 (apo-GIF/MT-3 plus one sulfur atom), and 7117 (oxidized GIF/MT-3 plus two sulfur atoms) suggested that oxidized apo-GIF/MT-3 contains sulfane sulfur-like species that presumably exist as intramolecular

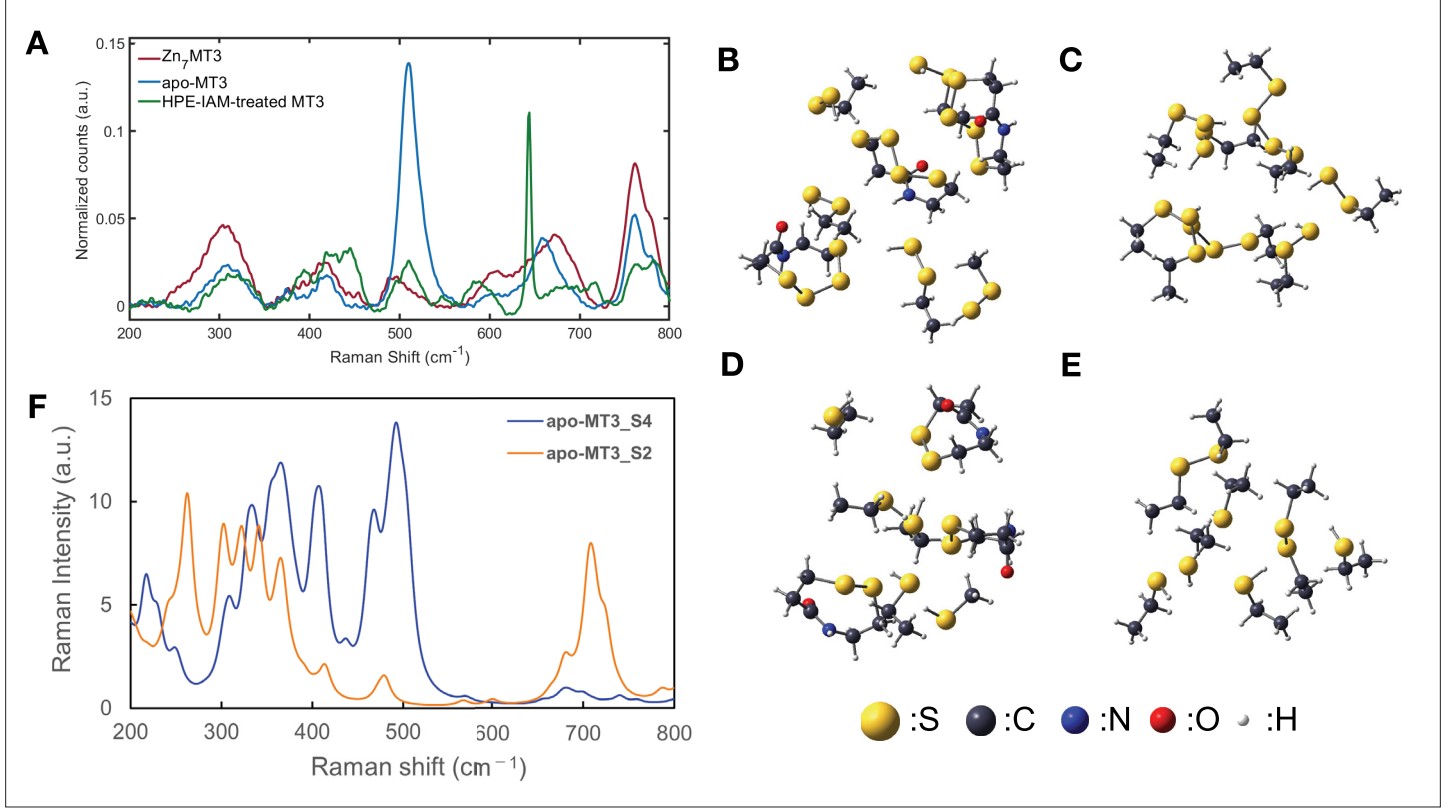

**Figure 2.** Detection of sulfane sulfur in GIF/MT-3 by Raman spectroscopy. (**A**) Raman spectra of Zn₇GIF/MT-3, oxidized apo-GIF/MT-3, and HPE-IAM-treated GIF/MT-3 in the 250–800 cm⁻¹ region. Optimized geometries for (**B**) α-domain and (**C**) β-domain models of apo-GIF/MT-3 (assuming some cysteines with persulfide and tetrasulfide bonds as shown). Optimized geometries for (**D**) α-domain and (**E**) β-domain models of apo-GIF/MT-3 (assuming some cysteines with disulfide bonds). (**F**) Calculated Raman spectra of apo-GIF/MT-3 models with/without sulfane sulfurs.

The online version of this article includes the following source data and figure supplement(s) for figure 2:

**Source data 1.** Peak assignments for apo-GIF/MT-3 model structures.

**Source data 2.** Peak assignments for Zn7S20GIF/MT-3 and Zn7GIF/MT-3 model structures.

**Source data 3.** Raw Raman spectroscopy data for panel A , and calculated Raman shifts and peak intensities for panel F .

**Figure supplement 1.** Zn-binding GIF/MT-3 models and calculated Raman spectra.

**Figure supplement 1—source data 1.** Calculated Raman spectroscopy data showing Raman shifts and peak intensities.

cysteine trisulfide or tetrasulfide bridges (*Figure 1D*). Note that an increase in mass of 32 Da can also result from addition of two oxygen atoms.

Raman spectroscopy is used to detect bonding changes in proteins, including MTs (*Torreggiani and Tinti, 2010*; *Devitt et al., 2018*). The Raman shift of Zn₇GIF/MT-3 (*Figure 2A*) contained a peak at 307 cm⁻¹, which is attributable to both S-terminal and S-bridging ligands (e.g. S–Zn, S–Zn–S) (*Torreggiani and Tinti, 2010*). Also, the peaks at 761 cm⁻¹ and 778 cm⁻¹ presumably corresponded to cysteine–metal bonds (e.g. C–S–Zn) and backbone vibrations, respectively (*Torreggiani and Tinti, 2010*). Overall loss of zinc was indicated by the decrease in the intensity of these peaks in the Raman spectra of apo-GIF/MT-3 and GIF/MT-3 treated with HPE-IAM to consume sulfane sulfur atoms (*Figure 2A*) and also confirmed using inductively coupled plasma (ICP)–MS (*Figure 1A*). The peak around 400 cm⁻¹ is reported to correspond to vibrations of metal–S bridges (e.g. Zn–S–Zn) (*Torreggiani and Tinti, 2010*); however, loss of such a peak was not observed for apo-GIF/MT-3 and HPE-IAM-treated GIF/MT-3. The Raman peaks within the 510–520 cm⁻¹ range reportedly indicate disulfide bonds in MT (*Torreggiani and Tinti, 2010*); in our spectra, an intense peak at 511 cm⁻¹ was observed for apo-GIF/MT-3 but not Zn₇GIF/MT-3 and HPE-IAM-treated GIF/MT-3 (*Figure 2A*).

In general, the S–S bonds of polysulfides are strong Raman scatterers due to the high polarizability of the bonding and lone pair electrons at the two-coordinate sulfur atoms (*Steudel and*

*Chivers, 2019*). It was also shown that S–S, S–S–S, and S–S–S–S structures exhibit different Raman shifts (*Schwab et al., 1979*; *Chivers and Lau, 1982*) and the different Raman bands reported for polysulfide may correspond to different geometrical forms of the molecule. To confirm the origin of the Raman peak at 511 cm$^{-1}$, quantum chemical calculations were made based on three-dimensional (3D) homology modeling, as described later, of apo-GIF/MT-3 structures with and without persulfides; the free cysteines and their persulfides were assumed to be protonated and some of the neighboring cysteines formed disulfides or tetrasulfides, depending on the model. The details of the model structures (*Figure 2B–E*) and the calculation schemes are described in the Materials and methods section. *Figure 2F* shows the calculated Raman spectra of apo-GIF/MT-3 with thiol (–SH) groups and disulfide (S–S) bonds (apo-GIF/MT-3_S2 model), and of apo-GIF/MT-3 persulfide (–SSH) groups and tetrasulfide (S–S–S–S) bonds (apo-GIF/MT-3_S4 model). Although the theoretical and experimental Raman spectra exhibited different overall shapes, owing to computational limitations, it is clear that the peak near 511 cm$^{-1}$ was markedly more intense for the apo-GIF/MT-3_S4 model than the apo-GIF/MT-3_S2 model. The normal mode vectors were evaluated, and the resulting assignments of these peaks are summarized in *Figure 2—source data 1*. The peaks mainly corresponded to the stretching and bending of disulfide and tetrasulfide bonds. The commensurate increase in peak intensity with the number of S–S bonds was consistent with the fact that the apo-GIF/MT-3_S4 model has several S–S and S–S–S–S bonds, while the apo-GIF/MT-3_S2 model has only S–S bonds. The Zn-binding models with/without persulfide also showed that the peaks around 500 cm$^{-1}$ were almost lost in Zn$_7$GIF/MT-3 without persulfide (*Figure 2—figure supplement 1* and *Figure 2—source data 2*), indicating that the peak near 488 cm$^{-1}$ (*Figure 2F*) for Zn$_7$GIF/MT-3 corresponded to the S–S structure of persulfide. Taken together, these results suggest the existence of sulfane sulfur atoms in both Zn$_7$GIF/MT-3 and apo-GIF/MT-3.

## Determination of sulfane sulfur atoms in Zn$_7$GIF/MT-3

HPE-IAM is a relatively inert electrophile that allows the detection of sulfur atoms (e.g. H$_2$S) by forming a bis-S-HPE-AM adduct (*Akaike et al., 2017*). Our rationale was that if GIF/MT-3 is an SSBP, the interaction of HPE-IAM with Zn$_7$GIF/MT-3 should eventually form a bis-S-HPE-AM adduct that can be quantified using LC–MS/MS with the stable isotope-labeled standard bis-S$^{34}$-HPE-AM (*Figure 3A*). Small molecules such as H$_2$S were removed during the purification of Zn$_7$GIF/MT-3 to exclude their contribution to the measured bis-S-HPE-AM adduct concentration. In a preliminary examination, a negligible amount of sulfane sulfur in Zn$_7$GIF/MT-3 could be detected after 36 hr incubation with HPE-IAM at 37°C. Stillman and coworkers reported that it was difficult for *N*-ethylmaleimide to access an apo-MT isoform at 37°C because of its folded structure, whereas heat treatment allowed such an electrophile to covalently bind the protein (*Irvine et al., 2018*). Therefore, as expected, the amount of sulfane sulfur detected in Zn$_7$GIF/MT-3 depended on the HPE-IAM concentration (plateauing at 5 mM), the amount of Zn$_7$GIF/MT-3 (up to 10 µM), and the reaction temperature and duration (*Figure 3B–D*). We performed the reaction with 5 mM HPE-IAM at 60°C for 36 hr. Under these optimized conditions, each MT isoform (each containing 20 cysteine residues) possessed approximately 20 sulfane sulfurs (*Figure 3E*). None of the Cys-to-Ala mutants of GIF/MT-3 possessed sulfane sulfur (*Figure 3E*), indicating that all 20 sulfane sulfurs were bound to cysteine residues of Zn$_7$GIF/MT-3. Although the form of binding (e.g. 20 RSSH, 10 RSSSH, RSS$_{20}$SR, 2 RSS$_{10}$SR) was not identified, persulfide (20 RSSH) was suggested to be formed rather than polysulfide for reasons described in the Discussion section.

Because this assay was performed at relatively high temperatures (60°C), we also examined the sulfane sulfur levels of several mutant proteins using chemically synthesized α- and β-domains of GIF/MT-3 to eliminate false-positive results. The amino acid sequences of the obtained recombinant proteins are shown in *Figure 3—figure supplement 1*. As shown in *Figure 3—figure supplement 2A*, sulfane sulfur (less than 1 molecule per protein) was undetectable in chemically synthesized α- and β-domains of GIF/MT-3, whereas several molecules of sulfane sulfur per protein were detected in recombinant α- and β-domains exhibited (*Figure 3—figure supplement 2B*, left panel). These findings indicated that the sulfane sulfur detected in our assay was derived from biological processes executed during the production of GIF/MT-3 protein. We further analyzed mutant proteins with β-Cys-to-Ala and α-Cys-to-Ala substitutions and found that their sulfane sulfur levels were comparable with those of the α- and β-domains of GIF/MT-3, respectively (*Figure 3—figure supplement 2B*, left panel). Additionally, the Ser-to-Ala mutation did not affect the sulfane sulfur levels of GIF/MT-3. The

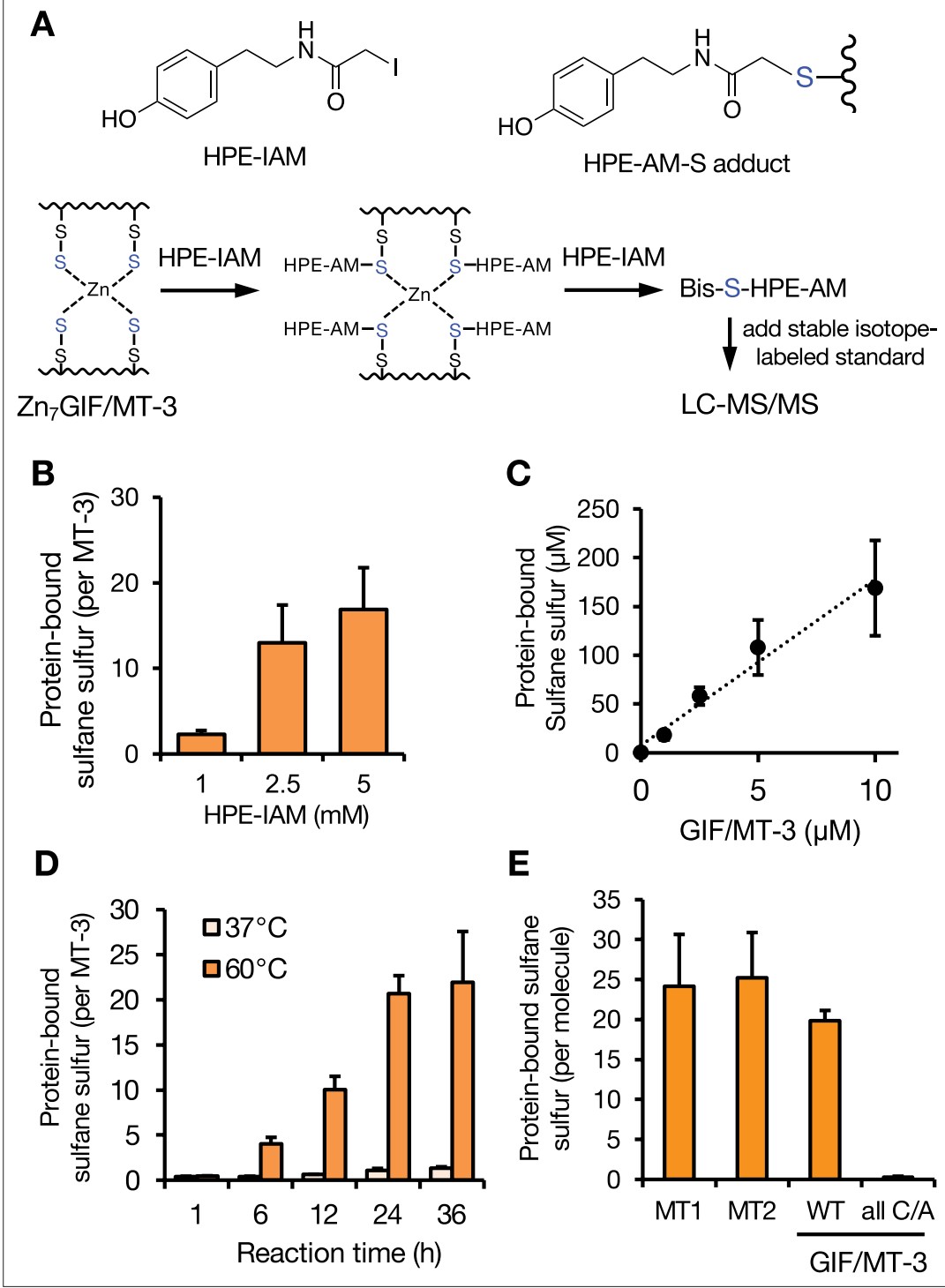

**Figure 3.** Sulfane sulfur assay optimization and quantification of MT sulfane sulfur content. (**A**) Schematic showing the detection of sulfane sulfur in $Zn_7GIF/MT$-3. (**B**) Sulfane sulfur detected in $Zn_7GIF/MT$-3 after incubation with the indicated concentrations of HPE-IAM at 60°C for 36 hr in 20 mM Tris-HCl (pH 7.5). (**C**) Sulfane sulfur detected in $Zn_7GIF/MT$-3 after incubation with 5 mM HPE-IAM at 60°C for 36 hr in 20 mM Tris-HCl (pH 7.5). (**D**) Sulfane sulfur detected in $Zn_7GIF/MT$-3 after incubation with 5 mM HPE-IAM at 37°C or 60°C for the indicated times in 20 mM Tris-HCl (pH 7.5). (**E**) Sulfane sulfur detected in human $Zn_7MT$-1, $Zn_7MT$-2, $Zn_7GIF/MT$-3, wild-type (WT) $Zn_7GIF/MT$-3, and apo-GIF/MT-3 with all Cys residues mutated to Ala (all C/A), each incubated with 5 mM HPE-IAM at 60°C for 36 hr in 20 mM Tris-HCl (pH 7.5). Sulfane sulfur content was measured using LC-MS/MS. HPE-IAM, β-(4-hydroxyphenyl)ethyl iodoacetamide. Each value represents the mean ± SD of three independent experiments.

*Figure 3 continued on next page*

*Figure 3 continued*

The online version of this article includes the following source data and figure supplement(s) for figure 3:

**Source data 1.** Source data for panels B-F: Protein-bound sulfane sulfur content measured in GIF/MT-3 samples.

**Figure supplement 1.** Amino acid sequences of human MTs and GIF/MT-3 mutant.

**Figure supplement 2.** Determination of sulfane sulfur and zinc content in wild-type and several mutant proteins.

**Figure supplement 2—source data 1.** Protein-bound sulfane sulfur and zinc content measured in GIF/MT-3 samples.

zinc content of each mutant protein was also determined under these conditions (*Figure 3—figure supplement 2B*, right panel).

## Redox-based GIF/MT-3 recycling system during oxidative stress

To explore the functional role of sulfane sulfur in GIF/MT-3, we examined the stability of sulfane sulfur in the protein with or without bound zinc. Freshly prepared $Zn_7GIF/MT-3$ and apo-GIF/MT-3 possessed almost the same amount of sulfane sulfur (*Figure 4A*). Unexpectedly, sulfane sulfur content in $Zn_7GIF/MT-3$ remained unchanged for up to 28 days in 20 mM Tris–HCl (pH 7.5) at 37°C (data not shown). In contrast, in apo-GIF/MT-3, sulfane sulfur content decreased markedly within 12 hr of incubation, and the addition of zinc blocked any further decrease (*Figure 4A*). This suggests that zinc ions are rapidly re-coordinated by the persulfide group in apo-GIF/MT-3, thereby stabilizing sulfane sulfur atoms. To examine the possibility of forming intramolecular cysteine tetrasulfide, which is stable and cannot react with iodoacetamide in the absence of a reducing agent (*Capdevila et al., 2021*), apo-GIF/MT-3 left in 20 mM Tris–HCl (pH 7.5) for 36 hr (i.e. oxidized apo-GIF/MT-3) was incubated with the reducing

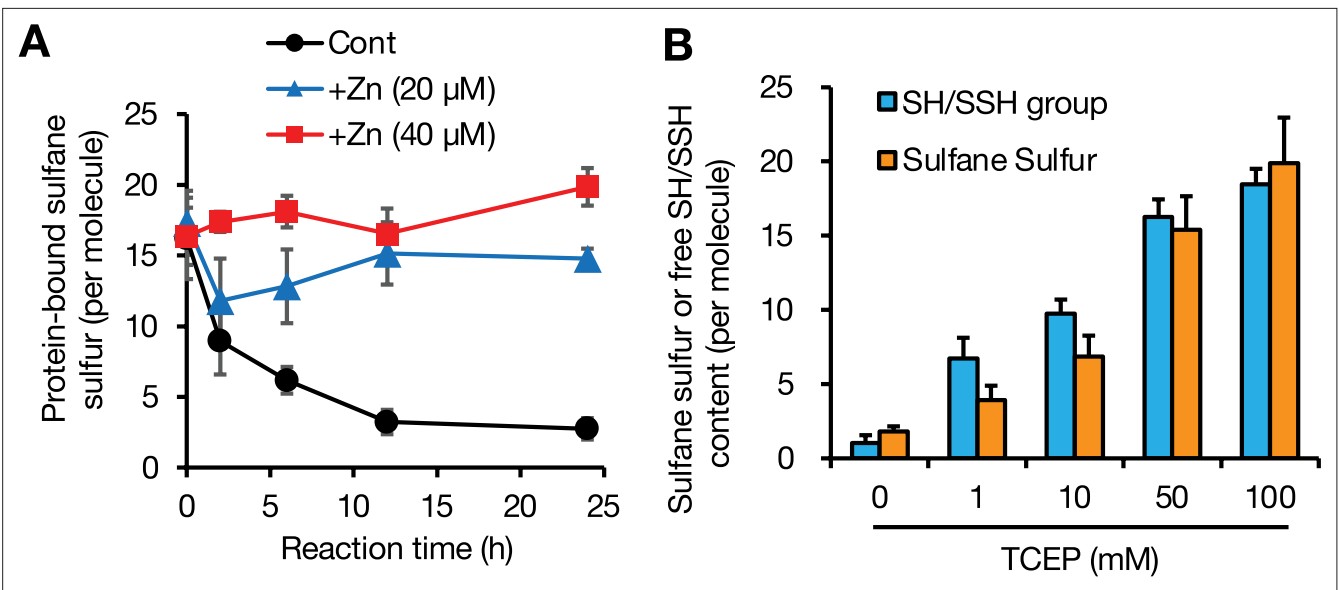

**Figure 4.** Sulfane sulfur stability in apo-GIF/MT-3 and its restoration by a reducing agent. (**A**) Stability of sulfane sulfur in apo-GIF/MT-3 incubated with or without (Cont) zinc. To prepare apo-GIF/MT-3, $Zn_7GIF/MT-3$ was incubated in 0.1 M HCl for 30 min, then the buffer was replaced with 20 mM Tris–HCl (pH 7.5). To examine the stability of sulfane sulfur in apo-GIF/MT-3, freshly prepared apo-GIF/MT-3 (2 µM) with or without added zinc ions was incubated at 37°C for up to 24 hr. (**B**) Effect of tris(2-carboxyethyl)phosphine (TCEP) on sulfane sulfur binding and free SH/SSH groups in oxidized apo-GIF/MT-3. To prepare oxidized apo-GIF/MT, $Zn_7GIF/MT-3$ was incubated in HCl (0.1 N) at 37°C for 30 min and then replaced with 20 mM Tris–HCl (pH 7.5) buffer and incubated for 36 hr at 37°C. The resulting oxidized apo-GIF/MT-3 protein (10 µM) was incubated with 0, 1, 10, 50, or 100 mM TCEP in 20 mM Tris–HCl (pH 7.5) at 37°C for 1 hr, then low-molecular-weight molecules were removed by 3 kDa ultrafiltration for six times. Sulfane sulfur content was determined using LC–ESI–MS/MS, and the concentrations of free SH/SSH groups were measured using Ellman's reagent. Each value represents the mean ± SD of three independent experiments.

The online version of this article includes the following source data and figure supplement(s) for figure 4:

**Source data 1.** Source data for panels A and B: Protein-bound sulfane sulfur and free SH/SSH content measured in GIF/MT-3 samples.

**Figure supplement 1.** A putative reaction scheme for DTNB with RSSH.

agent tris(2-carboxyethyl)phosphine (TCEP). The presence of free SH/SSH groups in oxidized apo-GIF/MT-3, determined using 5,5'-ditiobis-(2-nitrobenzoic acid) (DTNB) (*Ellman, 1959*), was negligible but increased following TCEP incubation, leading to a complete recovery of sulfane sulfur atoms (*Figure 4B*). These observations led us to assume that the time-dependent disappearance of persulfide in apo-GIF/MT-3 (*Figure 4A*) was not due to the oxidative degradation of sulfane sulfur but rapid closure of a ring that can be cleaved by TCEP. A putative reaction scheme for DTNB with RSSH is shown in *Figure 4—figure supplement 1*. Moreover, our method, based on TCEP-mediated reduction of tetrasulfide and subsequent trapping of sulfane sulfur atoms by HPE-IAM, was validated using the model compounds *N*-acetylcysteine-tetrasulfide (*Figure 5A*) and diallyltetrasulfide (*Figure 5B*). In the absence of TCEP, minimal amounts of bis-S-HPE-AM adducts were detected in all the compounds examined. However, incubation with TCEP resulted in the stoichiometric detection of sulfane sulfur but not oxidized-*N*-acetylcysteine and diallyl disulfide, suggesting that sulfane sulfur was stably trapped by HPE-IAM during the 36 hr TCEP incubation. A possible mechanism of reaction between tetrasulfide compounds and HPE-IAM is shown in *Figure 5C*. Thus, we confirmed that sulfane sulfur atoms of cysteine tetrasulfide in apo-GIF/MT-3 could be preserved after zinc release and oxidation.

Several reports have indicated that oxidative modification of MTs results in the release of zinc involved in zinc signaling (*Maret, 2019*; *Maret and Vallee, 1998*). Incubation with $H_2O_2$ and *S*-nitroso-*N*-acetylpenicillamine (SNAP), a nitrosonium ion donor, induced zinc release from $Zn_7$GIF/MT-3 (*Figure 6A*). Under these conditions, the numbers of free SH/SSH groups and sulfane sulfur atoms in GIF/MT-3 were also decreased by $H_2O_2$ and SNAP treatment. However, subsequent TCEP treatment nearly restored the original levels of free SH/SSH and sulfane sulfur (*Figure 6B*). The persulfide in apo-GIF/MT-3 appeared to be resistant to TCEP-induced release of sulfane sulfur, although tetrasulfide was not. Overall, it seems likely that sulfane sulfur in $Zn_7$GIF/MT-3 acts as a reserve of sulfur to be modified under physiological oxidative stress and a component of the redox-active closed-ring structure regulated by reductants (*Figure 6C*).

The interaction of sulfane sulfur species with KCN to yield thiocyanate and thiol products (cyanolysis) has been used as evidence of the presence of protein hydropersulfides (*Ono et al., 2014*). Therefore, we used KCN to eliminate the sulfane sulfur atoms from $Zn_7$GIF/MT-3. KCN treatment decreased the sulfane sulfur atom content of $Zn_7$GIF/MT-3 by approximately 75% (*Figure 7A*). After removing KCN, the reducing agent TCEP was subsequently added because intramolecular cysteine disulfide/tetrasulfide bridges can be formed under the condition. TCEP did not recover the level of sulfane sulfur in GIF/MT-3 (*Figure 7A*), indicating that KCN indeed removed sulfane sulfur from GIF/MT-3. In addition to eliminating the sulfane sulfur atoms, KCN treatment also reduced the zinc content of GIF/MT-3 (*Figure 7B*). To examine the role of sulfane sulfur in zinc retention, sulfane sulfur-diminished apo-GIF/MT-3 was incubated with zinc after TCEP treatment to reconstruct zinc-bound GIF/MT-3. Re-co-ordination of zinc ions to KCN-treated apo-GIF/MT-3 was incomplete compared with KCN-untreated apo-GIF/MT-3 (*Figure 7B*), implying the contribution of sulfane sulfur to zinc binding in GIF/MT-3.

## Reduction of apo-GIF/MT-3 by thioredoxin

Thioredoxin (Trx) is a master enzyme that reduces disulfide bonds in cellular proteins (*Lu and Holmgren, 2014*). Holmgren previously reported that *E. coli* Trx predominantly catalyzes S–S bond cleavage of insulin ($K_m$ ~ µM) rather than low-molecular-weight substances such as cystine and oxidized GSH (*Holmgren, 1979*). Surprisingly, apo-GIF/MT-3 ($K_m$ = 30 nM, $K_{cat}$ = 31,536 min$^{-1}$, $K_{cat}/K_m$ = 1051 × $10^6$ M$^{-1}$min$^{-1}$) was a much more efficient substrate than insulin ($K_m$ = 1,192 nM, $K_{cat}$ = 19,114 min$^{-1}$, $K_{cat}/K_m$ = 16 × $10^6$ M$^{-1}$min$^{-1}$) (*Figure 8A and B*), whereas Trx was unable to reduce $Zn_7$GIF/MT-3 (*Figure 8C*), as was the case for TCEP (*Figure 7A*). Therefore, we hypothesized that the zinc/persulfide clusters in $Zn_7$GIF/MT-3 may block the interaction of the protein with Trx and that the zinc ions bound to GIF/MT-3 may act as a repressor of Trx-mediated S–S bond cleavage. In addition, apo-GIF/MT-3 was a poor substrate for Trx-related proteins 14 (TRP14) and 32 (TRP32) (*Figure 8C*). Furthermore, HPE-IAM trapping assay analysis confirmed that the Trx/Trx reductase (TR) system recovered sulfane sulfur content in apo-GIF/MT-3 (*Figure 8D*) but not in the low-molecular-weight (<3 kDa) fraction (data not shown). These observations indicate that Trx is a powerful enzyme that cleaves the tetrasulfide bond in apo-GIF/MT-3 and that persulfide formation derived from apo-GIF/MT-3 seems to be resistant to Trx, as well as TCEP. Notably, a small amount of sulfane sulfur in apo-GIF/MT-3 was observed even in the absence of Trx and TR, thereby supporting our conclusion (*Figure 2B–E*) that

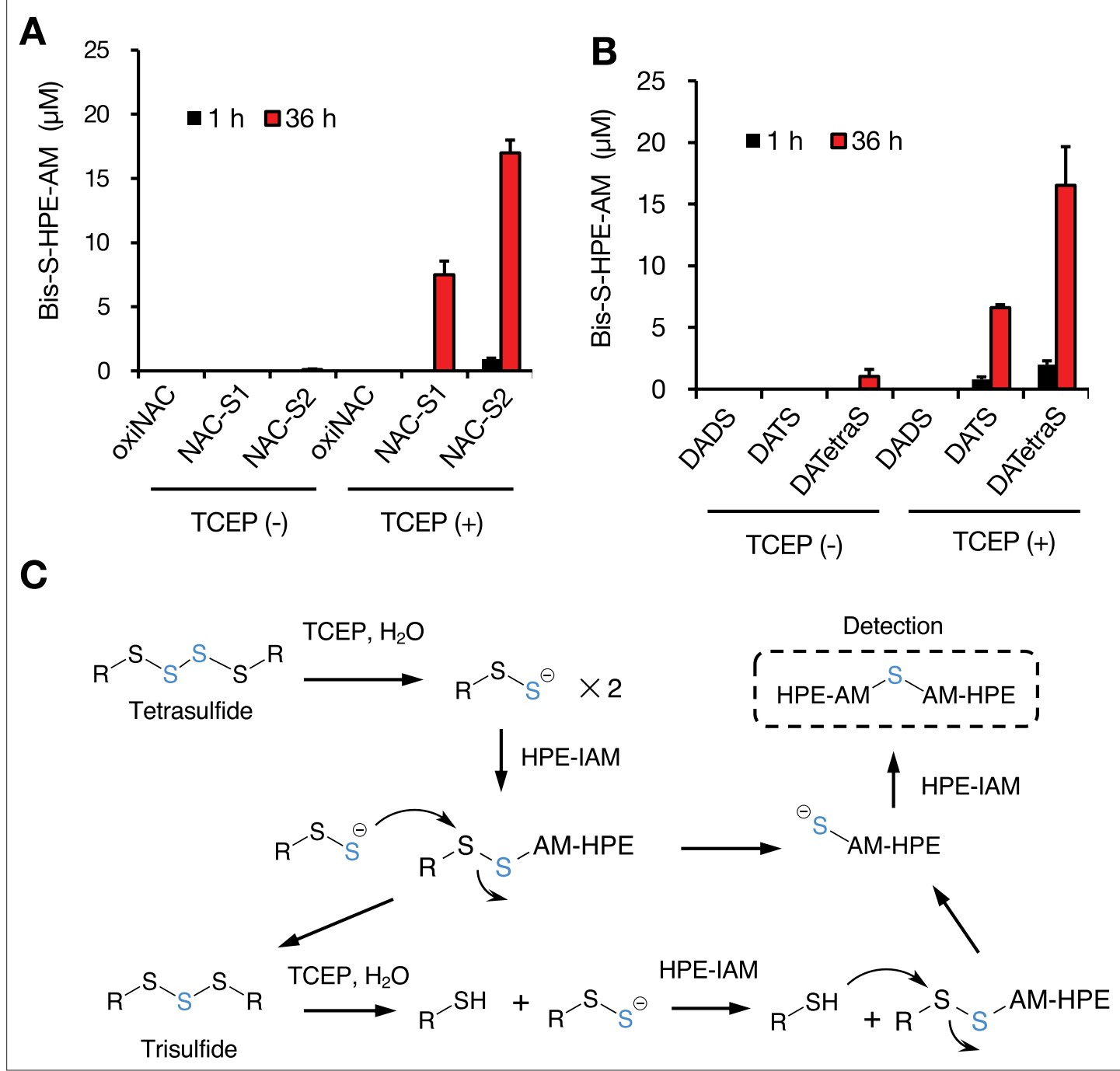

**Figure 5.** Reactivity of HPE-IAM with tetrasulfide derivatives as models of tetrasulfide bridges in apo-GIF/MT-3. (**A**) Reactivity of HPE-IAM with *N*-acetylcysteine (NAC) derivatives. Oxidized NAC (oxiNAC), NAC-trisulfide (NAC-S1), and NAC-tetrasulfide (NAC-S2) (each 10 µM) were incubated with HPE-IAM (5 mM) at 60°C for 1 or 36 hr with or without TCEP (1 mM) in 100 mM Tris-HCl (pH 7.5). (**B**) Reactivity of HPE-IAM with diallyl polysulfide derivatives. Diallyl disulfide (DADS), diallyl trisulfide (DATS), or diallyl tetrasulfide (DATetraS) (each 10 µM) was incubated with HPE-IAM (5 mM) at 60°C for 1 or 36 hr with or without TCEP (1 mM) in 100 mM Tris-HCl (pH 7.5). (**C**) Scheme showing possible reactions of tetrasulfide derivatives with HPE-IAM and TCEP. Bis-S-HPE-AM, bis-S-β-(4-hydroxyphenyl)ethyl acetamide. Each value represents the mean ± SD of three independent experiments.

The online version of this article includes the following source data for figure 5:

**Source data 1.** Source data for panels A and B: Measured Bis-S-HPE-AM concentrations in polysulfide derivatives after HPE-AM incubation.

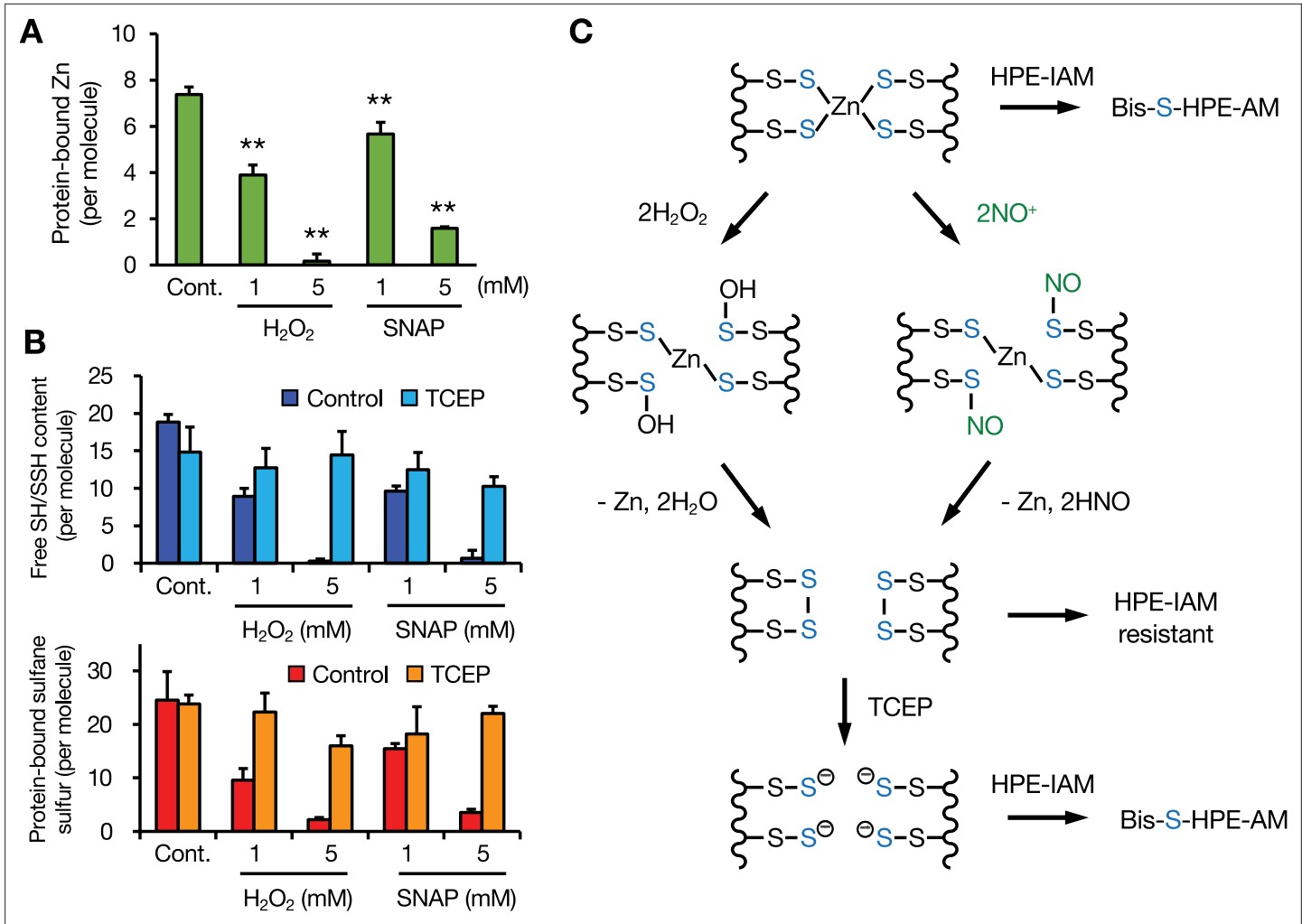

**Figure 6.** Redox-dependent release of zinc ions and recycling of sulfane sulfur in GIF/MT-3. (**A**) Quantitation of zinc ions released from $Zn_7$GIF/MT-3 by $H_2O_2$ and $S$-nitroso-$N$-acetylpenicillamine (SNAP). To examine the release of zinc ions by $H_2O_2$ and SNAP, $Zn_7$GIF/MT-3 (10 µM) was treated with $H_2O_2$ (1 or 5 mM) or SNAP (1 or 5 mM) in 100 mM Tris–HCl (pH 7.5) at 25°C for 30 min. After removing $H_2O_2$/SNAP using 3 kDa ultrafiltration four times, free SH/SSH groups and sulfane sulfur content in GIF/MT-3 were determined. (**B**) Free SH/SSH content in $Zn_7$GIF/MT-3, determined by $H_2O_2$ or SNAP treatment after incubation with TCEP. To examine the interaction of $Zn_7$GIF/MT-3 with $H_2O_2$ or NO, $Zn_7$GIF/MT-3 (10 µM) was incubated with $H_2O_2$ (1 or 5 mM) or SNAP (1 or 5 mM) in 100 mM Tris–HCl (pH 7.5) at 25°C for 30 min. After removing $H_2O_2$/SNAP using 3 kDa ultrafiltration four times, the resulting proteins (5 µM) were incubated with TCEP (50 mM) in 100 mM Tris–HCl (pH 7.5) at 37°C for 1 hr. After removing TCEP using 3 kDa ultrafiltration for five times, sulfane sulfur content was determined using LC–ESI–MS/MS and the concentrations of free SH/SSH groups were measured using Ellman's reagent. (**C**) Proposed reactions between a zinc/persulfide cluster in GIF/MT-3 and $H_2O_2$ or NO. Each value represents the mean ± SD of three independent experiments.

The online version of this article includes the following source data for figure 6:

**Source data 1.** Measured protein-bound zinc, sulfane sulfur, and free SH/SSH in GIF/MT-3 samples.

some persulfides or their deprotonated forms exist in apo-GIF/MT-3. This may be a unique SSBP-related feature of GIF/MT-3.

## 3D modeling of GIF/MT-3 with sulfane sulfur atoms

We generated a 3D homology model of human $Zn_7$GIF/MT-3 using the Molecular Operating Environment (MOE) software and the Protein Data Bank (PDB) structures rat MT-2 (4MT2) and α-domain of human MT-3 (2F5H) as templates (*Figure 9—figure supplement 1*). Then, we created a 3D structure of sulfane sulfur-bound $Zn_7$GIF/MT-3 (one sulfane sulfur per cysteine residue), $Zn_7S_{20}$GIF/MT-3 (*Figure 9A*), which was also used for Raman spectra modeling (*Figure 2B–E*). The predicted 3D structure of $Zn_7S_{20}$GIF/MT-3 was almost the same as that of $Zn_7$GIF/MT-3 (*Figure 9A*), with the root-mean-square

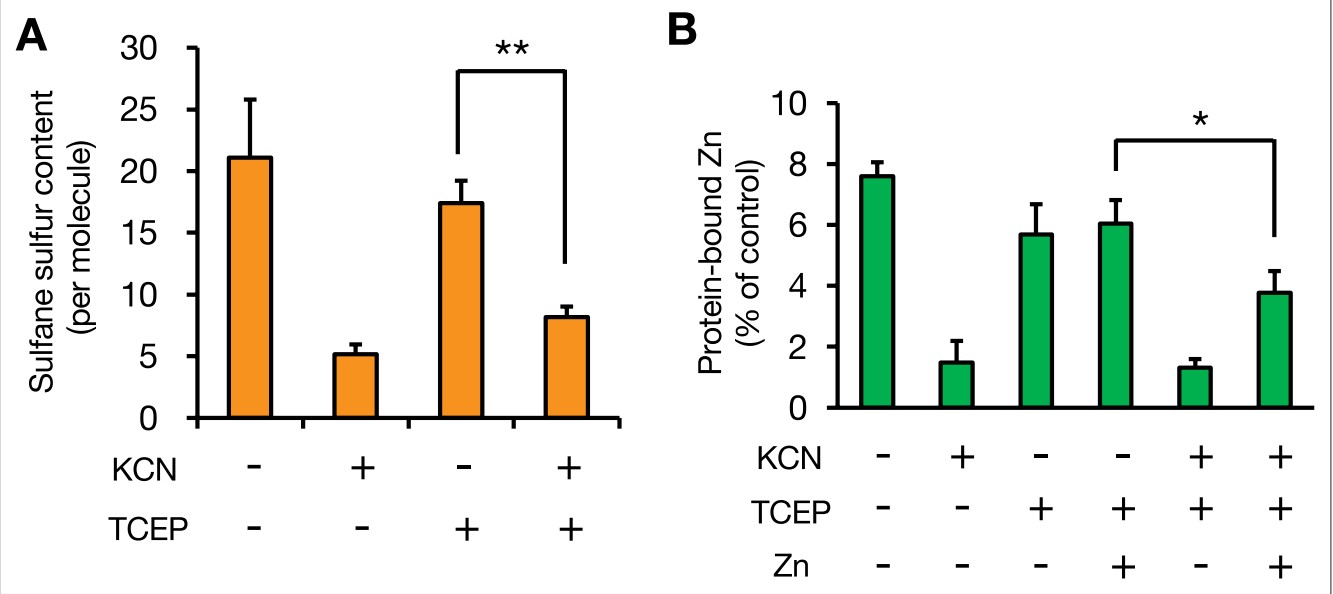

**Figure 7.** Contribution of sulfane sulfur in GIF/MT-3 to zinc binding. (**A**) To eliminate sulfane sulfur in $Zn_7GIF/MT$-3 by cyanolysis, $Zn_7GIF/MT$-3 (10 μM) was reacted with KCN (75 mM) in 100 mM Tris–HCl (pH 7.5) at 37°C for 14 hr. After removal of KCN, the resulting protein was incubated with TCEP (10 mM) in 100 mM Tris–HCl (pH 7.5) at 37°C for 1 hr. After removal of TCEP, sulfane sulfur content in GIF/MT-3 was determined using LC–ESI–MS/MS. (**B**) Comparison of zinc-binding capacity of GIF/MT-3 before and after cyanolysis. $Zn_7GIF/MT$-3 (10 μM) was incubated with KCN (75 mM) in 100 mM Tris–HCl (pH 7.5) at 37°C for 14 hr. After removal of KCN, the resulting protein was incubated with TCEP (10 mM) in 100 mM Tris–HCl (pH 7.5) at 37°C for 1 hr. After removal of TCEP, the resulting protein (5 μM) was incubated with zinc chloride (50 μM) in 50 mM Tris–HCl (pH 7.5) at 37°C for 1 hr. Low-molecular-weight molecules were removed using 3 kDa ultrafiltration after each step. Protein-bound zinc content was determined using ICP–MS. *p<0.05 and **p<0.01. Each value represents the mean ± SD of three independent experiments.

The online version of this article includes the following source data for figure 7:

**Source data 1.** Protein-bound sulfane sulfur and zinc content measured in GIF/MT-3 samples.

deviation of atom positions being only 0.789 Å. Similar results were obtained for $Zn_7MT1$ and $Zn_7MT2$ (**Figure 9—figure supplement 2**). $Zn_7GIF/MT$-3 contains a cyclohexane-like $Zn_3Cys_9$ cluster in the β-domain and a bicyclononane-like $Zn_4Cys_{11}$ cluster in the α-domain (**Vasák, 2005**). The structures of both clusters were mostly maintained even when all thiol groups were changed to persulfides by adding one sulfane sulfur atom to each (**Figure 9B**). Schematic structures of the generated $Zn_7GIF$/MT-3 with or without sulfane sulfurs are shown in **Figure 9—figure supplement 3**. Addition of two sulfane sulfur atoms, corresponding to cysteine trisulfide, disrupted each cluster structure (**Figure 9—figure supplement 4**). When they contained one sulfane sulfur in each cysteine, the thermostability scores of MT1, MT2, and GIF/MT-3 in the presence of zinc ions were higher than those in the absence of zinc ions (**Figure 9—source data 1**), indicating that these ions are key to the thermostability of MT isoforms, including those containing sulfane sulfur atoms. The thermostabilities and $Cd_7$, $Cu_7$, $Hg_7$, and $Zn_7$ binding affinities of $S_{20}GIF/MT$-3 were more favorable than the corresponding values of the sulfane sulfur-free protein (**Table 1**). Conversely, placing more than one sulfane sulfur on each cysteine residue decreased the thermostability and zinc-binding affinity (**Figure 9C**). Collectively, these results indicate that zinc ions contribute to protection against persulfide oxidation and MT thermostability, while sulfane sulfur atoms participate in cysteine tetrasulfide formation and enhancement of metal-binding affinity. Therefore, our study provides evidence for an interdependence of zinc and sulfane sulfur and for unique structural and functional roles of the persulfide groups in GIF/MT-3.

To confirm the presence of SSBPs in mouse brain, we attempted to isolate them from the high-molecular-weight fraction (>3 kDa) of the cytosol using diethylaminoethyl (DEAE)-Sepharose CL-6B column chromatography. Surprisingly, an abundance of SSBPs was detected, and approximately half of them were tightly bound to the column and eluted in buffer containing 0–0.4 M NaCl (**Figure 10A**). Although SSBPs from other proteins with iron/sulfur clusters may also have been detected, this possibility remains to be explored in future studies. Eluate corresponding to the peak concentration of SSBPs (fractions 40–44) was further separated using Sephacryl S-100 column chromatography

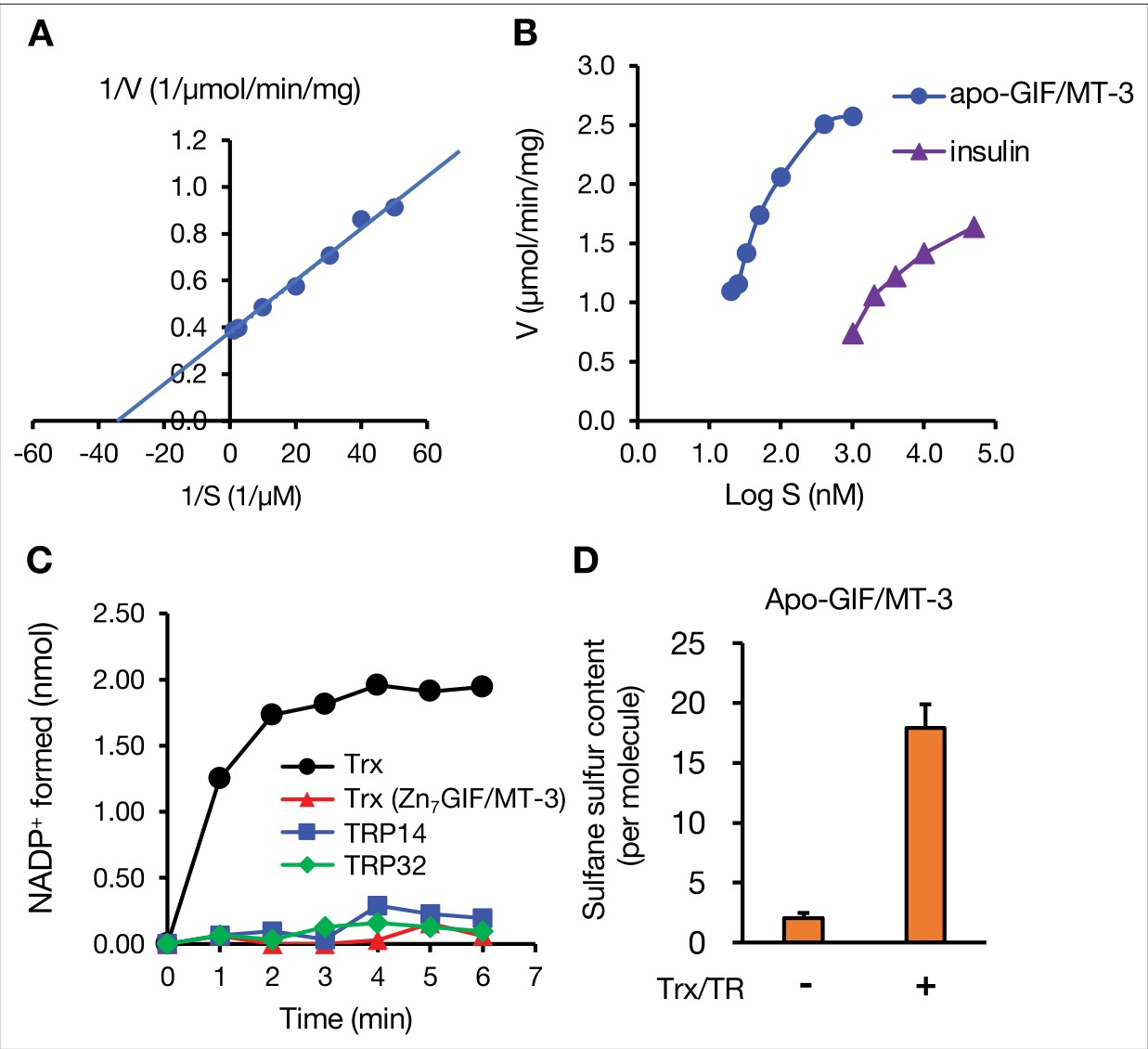

**Figure 8.** Reduction of apo-GIF/MT-3 by thioredoxin (Trx) and subsequent regeneration of sulfane sulfur. (**A**) Velocity (V) of Trx-catalyzed reduction of oxidized apo-GIF/MT-3 substrate (S). Oxidation of NADPH was followed by measuring the absorbance of NADPH at 340 nm. (**B**) Comparison of substrate reduction by NADPH and the Trx system. (**C**) NADP+ formation upon incubation of: oxidized apo-GIF/MT-3 with Trx/TR, TRP14/TR, or TRP32/TR; and $Zn_7$GIF/MT-3 with Trx/TR. (**D**) Regeneration of sulfane sulfur in oxidized apo-GIF/MT-3 after incubation with the Trx/TR system. TR, Trx reductase; TRP14, Trx-related protein 14; TRP32, Trx-related protein 32. Representative data are shown. Similar results were obtained in at least two independent experiments. For panel D, each value represents the mean ± SD of three independent experiments.

The online version of this article includes the following source data for figure 8:

**Source data 1.** NADPH oxidation over time and protein-bound sulfane sulfur at endpoint during incubation of GIF/MT-3 with the Trx/TR system.

(*Figure 10B*), which resolved two major SSBP-related peaks. We collected eluate corresponding to the latter (fractions 40–43), which contained proteins with high SSBP content and a mass of approximately 13 kDa, even though its total protein concentration was low. These proteins were then separated using Blue Sepharose chromatography. While proteins that bound tightly to the Blue Sepharose resin did not contain sulfane sulfur, an SSBP that eluted in fractions 3–5 migrated as a single band (16 kDa) using SDS-PAGE (data not shown). This 16 kDa SSBP was confirmed to be GIF/MT-3 using nano ultra-performance LC–MS/MS (*Figure 10—source data 1*).

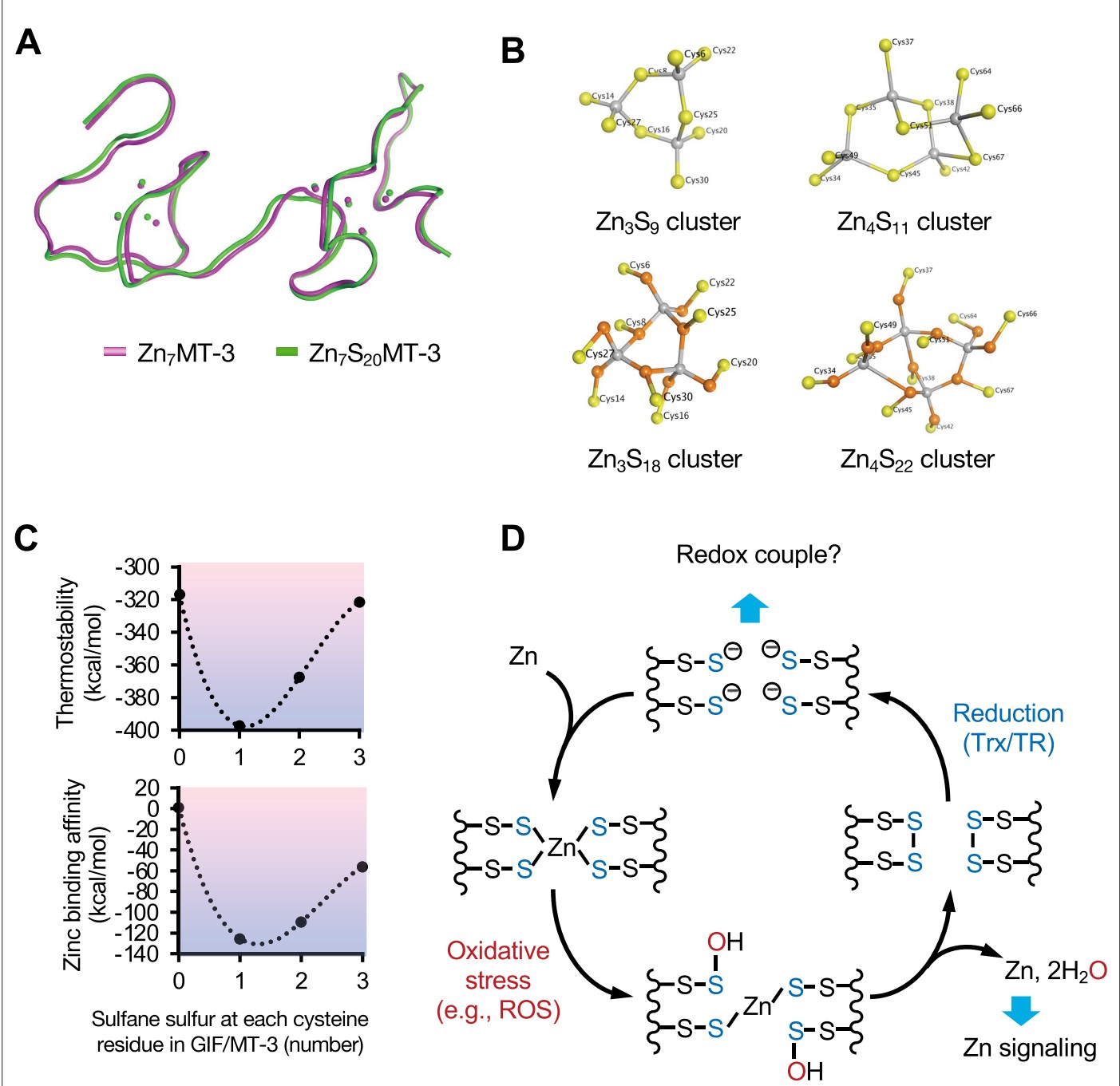

**Figure 9.** Structural modeling of sulfane sulfur in GIF/MT-3 using MOE, and a reaction scheme for sulfane sulfur-based zinc/persulfide cluster. (**A**) Comparison of three-dimensional structures of $Zn_7$GIF/MT-3 (pink) and $Zn_7S_{20}$GIF/MT-3 (green). (**B**) Cyclohexane-like $Zn_3Cys_9$ cluster in the GIF/MT-3 homology model, and bicyclononane-like $Zn_4Cys_{11}$ cluster derived from PDB structure 2F5H with (lower) or without (upper) sulfane sulfur. Yellow, orange, and gray spheres indicate cysteine residues, sulfane sulfur, and zinc ions, respectively. (**C**) Thermostability and zinc-binding affinity scores of GIF/MT-3 with different numbers of sulfane sulfurs at each cysteine residue. (**D**) A proposed model for redox-dependent hold-and-release regulation of zinc ions by GIF/MT-3.

The online version of this article includes the following source data and figure supplement(s) for figure 9:

**Source data 1.** Thermostability score of sulfane sulfur-bound MT isoforms with or without Zn.

**Source data 2.** Source data for panel C: Thermostability and zinc binding affinity scores of GIF/MT-3 variants with varying sulfane sulfur counts at each cysteine.

**Figure supplement 1.** Homology modeling of GIF/MT-3.

*Figure 9 continued on next page*

*Figure 9 continued*

**Figure supplement 2.** 3D structural models of human $Zn_7$MT-1A and $Zn_7$MT-2.

**Figure supplement 3.** Schematic structures of $Zn_7$GIF/MT-3 (**A**) and $Zn_7S_{20}$GIF/MT-3 (**B**).

**Figure supplement 4.** $Zn_3Cys_9$ and $Zn_4Cys_{11}$ clusters containing the polysulfide form of sulfane sulfur.

## Discussion

MT isoforms are believed to possess a zinc/thiolate cluster characterized by strong Zn–S bonds but facile release of zinc ions occurs under oxidative stress (*Kang, 2006*; *Maret, 2008*). Here, we developed a reliable assay involving prolonged (36 hr) incubation with HPE-IAM at 60°C to extract sulfane sulfur from $Zn_7$GIF/MT-3 and isotope-dilution LC–MS/MS analysis (*Figure 3*). The present study provided evidence that MTs are SSBPs containing a sulfane sulfur atom on each of their 20 cysteine residues, which form a zinc/persulfide cluster. Although Capdevila et al. previously showed evidence for the existence of sulfide ions in recombinant MT1, MT2, and MT4, they detected only one to four sulfide ions liberated from MTs as $H_2S$ gas using gas chromatography–flame photometric detection in strongly acidic conditions (*Capdevila et al., 2005*). These observations suggest that, unlike our assay, this method is likely to underestimate the sulfane sulfur content of proteins. While our assay has difficulty detecting sulfane sulfur in oxidized polysulfide bridges such as cysteine tetrasulfide, which is supported by a recent observation (*Capdevila et al., 2021*), a TCEP cleavage step enabled this problem to be overcome (*Figure 5C*). However, FT–ICR–MALDI–TOF/MS analysis failed to detect sulfur modifications in GIF/MT-3 (*Figure 1B*), suggesting that sulfur modifications in the protein were dissociated during laser desorption/ionization. Therefore, we postulate that the small amount of sulfur detected in oxidized apo-GIF/MT-3 is derived from the effect of laser desorption/ionization rather than any actual modification of the minority component.

Sulfane sulfur modification of MTs appears to be universal because similar sulfane sulfur contents were observed in recombinant human MT-1, MT-2, and GIF/MT-3 (*Figure 3E*). Although MTs have been mainly studied in vertebrates, their diversity and distribution have been widely reported (*Ziller and Fraissinet-Tachet, 2018*). So far, three functional MT forms (reduced apoprotein, oxidized apoprotein, and metalated protein) are known. In this current study, we propose that sulfane sulfur is another key factor involved in regulation of MT function that may have major implications for several biological functions.

Oxidation of sulfane sulfur in GIF/MT-3 is a fascinating mechanism of zinc release that does not involve direct thiol oxidation. There is little doubt that zinc release from the zinc/persulfide cluster in GIF/MT-3 will be sensitive to mild oxidative stress because the $pK_a$ value of cysteine persulfide is lower than that of cysteine (*Cuevasanta et al., 2015*). However, the formation of cysteine tetrasulfides following zinc release from the oxidized GIF/MT-3 (*Figure 9D*) represents a paradigm shift in GIF/MT-3 biochemistry. The sulfane sulfur atoms of MT–tetrasulfide are stably retained and can reacquire zinc after they are reduced, as shown in *Figure 9D*. Tetrasulfides, whose presence in apo-GIF/MT-3 was shown (*Figures 1C and 2A*), are presumably formed in response to structural frustration of the disulfide form, as reported for the sulfide-responsive transcriptional repressor, SqrR (*Capdevila et al., 2021*). In our preliminary study, we failed to directly identify cysteine tetrasulfide during the reaction

**Table 1.** Thermostability and metal-binding affinity scores of growth inhibitory factor (GIF)/metallothionein-3 (MT-3) with or without sulfane sulfur.

Values were calculated using the Protein Design module of the Molecular Operating Environment (MOE) software.

| Metal | Sulfane sulfur | Affinity (kcal/mol) | Stability (kcal/mol) |
|---|---|---|---|
| | 0 | 11 | −302 |
| $Zn_7$ | 20 | −154 | −407 |
| | 0 | 12 | −282 |
| $Cd_7$ | 20 | −82 | −344 |
| | 0 | 13 | −276 |
| $Hg_7$ | 20 | −112 | −354 |

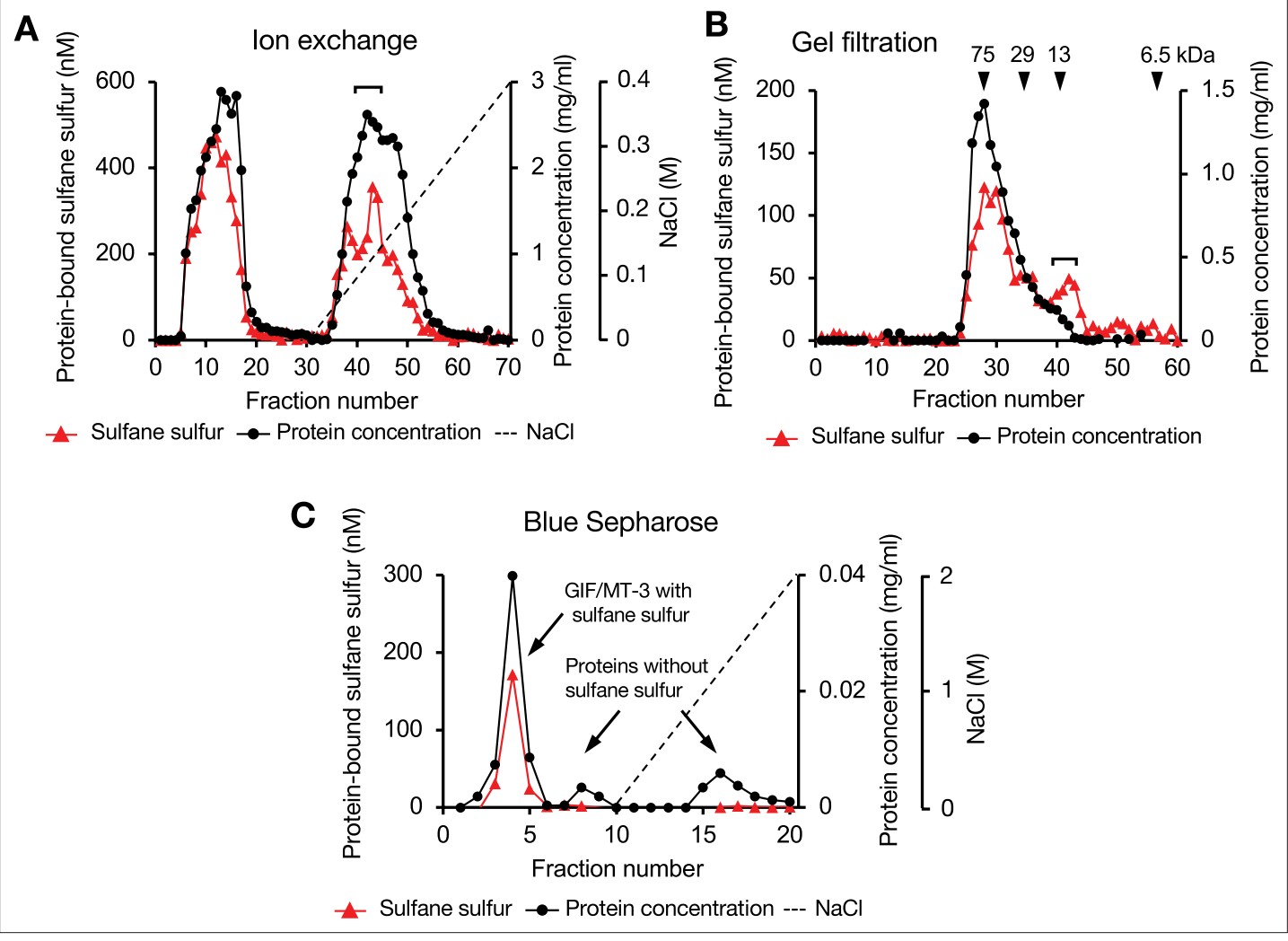

**Figure 10.** Separation of sulfane sulfur-binding proteins from mouse brain cytosol using column chromatography. (**A**) Diethylaminoethyl Sepharose CL-6B column. (**B**) Sephacryl S-100 column. (**C**) Blue Sepharose column. Triangles, closed circles, and dotted lines indicate sulfane sulfur, protein, and NaCl concentrations, respectively. Portions of each fraction were incubated with 5 mM of HPE-IAM in 20 mM Tris (pH 7.5) at 37°C for 30 min and the sulfane sulfur content was determined from the bis-S-HPE-AM adduct concentration measured using LC–MS/MS. Protein concentration was determined using the bicinchoninic acid assay. Isolation of sulfane sulfur-binding protein was performed as described in the Experimental procedures.

The online version of this article includes the following source data for figure 10:

**Source data 1.** Fragment sequences of a mouse brain sulfane sulfur-binding protein, determined using nano-UPLC–MS.

**Source data 2.** Source data for panels A-C: Protein-bound sulfane sulfur and protein concentrations measured in each fraction.

of apo-GIF/MT-3 with pronase because alkaline hydrolysis of polysulfide makes it unstable in water (*Hamid et al., 2019*; *Sawa et al., 2022*). We therefore speculate that cysteine tetrasulfide groups exist in an acidic local environment of apo-GIF/MT-3 and that Trx rapidly interacts with and cleaves tetrasulfide S–S bonds because of their markedly low p$K_a$ values, resulting in the regeneration of persulfide groups.

It is believed that the function of the sulfane sulfur moiety in proteins is to protect cysteine thiols from irreversible oxidation (e.g. to sulfinic acid and sulfonic acid) because such oxidative modifications of cysteine persulfides can be reversed by Trx-mediated reduction (*Filipovic et al., 2018*). In other words, sulfane sulfurs act as 'sacrificial' sulfur atoms. However, in this study, we discovered that sulfane sulfur has a higher affinity for zinc than thiol (*Figure 9C* and *Table 1*) and is stored in non-sacrificial tetrasulfide groups when oxidized (*Figure 9D*). As a result, tetrasulfides in oxidized-apo-GIF/MT-3 are reduced by the Trx system, with Trx being regenerated to a reduced form by NADPH and TR, thereby enabling reacquisition of zinc ions by the persulfide moiety. This suggests that the persulfide moiety

in GIF/MT-3 appears to be relatively stable against Trx reduction. In contrast, Trx has been proposed to reduce the persulfide moiety of PTP1B (*Krishnan et al., 2011*) and albumin (*Dóka et al., 2016*; *Wedmann et al., 2016*). A possible explanation for this discrepancy is that apo-GIF/MT-3-persulfide is rapidly changed into a different conformation that is topologically resistant to Trx reduction. In other words, Trx may exhibit substrate specificity.

While our proposed mechanism of MT redox regulation is consistent with that proposed by *Maret, 2008*, discovery of sulfane sulfur in MTs explains unknown features of MT redox biochemistry. Maret and coworkers showed that approximately 50% of MT existed in the apo form in the rat brain, although they did not examine specific isoforms (*Yang et al., 2001*). A reasonable explanation for this observation is that reduced apo-MT can undergo redox-coupled reactions with oxidized proteins (*Sagher et al., 2006*), leading to the formation of apo-MT-tetrasulfide, which loses its zinc ion-binding capability. GIF/MT-3 is a constitutive form predominantly expressed in the brain (*Vašák and Meloni, 2017*) and protects against Alzheimer's disease (*Uchida, 2010*; *Koh and Lee, 2020*). Notably, several studies have indicated that Trx acts as a protection factor in Alzheimer's disease (*Jia et al., 2021*; *Akterin et al., 2006*). Although the involvement of GIF/MT-3 and Trx in regulating this disorder has not been elucidated, Trx-mediated reduction of apo-GIF/MT-3 may lead to the reduction of unknown proteins that might suppress the onset of Alzheimer's disease because of the high antioxidant capabilities of apo-MT persulfides. Further studies are required to identify the redox-coupled protein dynamics associated with the reduced form of apo-GIF/MT-3 and their relevance to the molecular basis of Alzheimer's disease.

To our knowledge, this is the first study to perform 3D structural modeling of an SSBP with or without sulfane sulfur. Our structural modeling studies revealed that, like $Zn_7$GIF/MT-3, the seven zinc ions of $Zn_7S_{20}$GIF/MT-3 were tetrahedrally coordinated by the array of 20 sulfane sulfurs but not by cysteine thiols, thereby forming two zinc/persulfide clusters (*Figure 9B*). As shown in *Figure 9A*, *Figure 9—figure supplement 2*, cyclohexane-like $Zn_3S_9$ and bicyclononane-like $Zn_4S_{11}$ clusters were conserved even in the presence of 20 sulfane sulfurs without affecting the overall structure of the MTs. Nevertheless, sulfane sulfur is important for both metal-binding affinity and protein stability (*Table 1*) and thus MT function. We suggest that the 20 sulfane sulfurs are evenly distributed among 20 cysteine residues in GIF/MT-3 for the following reasons: (i) zinc binding of $Zn_7$GIF/MT-3 was suggested to involve a C–S–S–Zn rather than Zn–S–Zn structure (*Figure 2A*); (ii) addition of two or three sulfane sulfurs to each cysteine residue in $Zn_7$GIF/MT-3 decreased its stability (*Figure 9C*); (iii) the polysulfide form of sulfane sulfur seems to have difficulty maintaining zinc/sulfur clusters in GIF/MT-3 (*Figure 9—figure supplement 4*). However, one of the limitations of our study is that we did not directly observe such zinc–persulfide cluster itself.

Because protein sulfuration occurs during nascent protein translation, SSBPs appear to be ubiquitous in cells. In fact, we previously reported that a variety of cellular proteins were per/poly-sulfidated, as determined using a tag-switch-tag assay (*Ida et al., 2014*). We have recently characterized several SSBPs such as GSH S-transferase P1 (GSTP1) (*Abiko et al., 2015*), dynamin-related protein 1 (Drp1), alcohol dehydrogenase 5 (ADH5), glyceraldehyde-3-phosphate dehydrogenase (GAPDH), ethylmalonic encephalopathy 1 (ETHE1) (*Akaike et al., 2017*), and now MT-1, MT-2, and GIF/MT-3. As shown in *Figure 10*, the present assay also confirmed that there are numerous SSBPs in addition to GIF/MT-3 in the cytosol of the mouse brain. Notably, 3207 zinc-binding proteins in the human proteome have been identified with three and four zinc ligands, with 40% of the latter category consisting of Cys4 coordination (*Andreini et al., 2006*). Once oxidants react with thiols in a zinc finger domain, zinc is released from the coordination site, resulting in the inhibition of the zinc finger protein function. However, disulfide formation in the zinc finger domain is reversed by reducing agents such as GSH. We speculate that sulfane sulfur is a key component of such a 'zinc redox switch', enabling it to provide high affinity for zinc and protection against excess oxidative/electrophilic stress. Supporting our notion, zinc finger proteins such as tristetraprolin (*Lange et al., 2019*), androgen receptor (*Zhao et al., 2014*), and prolyl hydroxylase (*Dey et al., 2020*) were reported to undergo persulfidation by $H_2S$. In this context, $H_2S$ inhibits enzymatic activity of tristetraprolin and androgen receptor, but increases enzymatic activity of prolyl hydroxylase. A possible explanation for this difference is that excess exogenous $H_2S$ may cause protein polysulfidation, thereby disturbing the native protein structure. Our findings provide structural and mechanistic insights into the role of sulfane sulfur in hold-and-release regulation of zinc ions by zinc-binding

proteins. A future research goal is to investigate the universal role of sulfane sulfur atoms in redox regulation of all zinc-finger proteins.

In conclusion, we have provided evidence that S-sulfuration based on the addition of sulfane sulfur plays a central role in hold-and-release regulation of zinc by GIF/MT-3. Our study has further revealed a fascinating redox-dependent switching mechanism of a zinc/persulfide cluster involving the formation of a cystine tetrasulfide bridge. The biological significance of sulfane sulfur in MTs lies in its ability to (1) contribute to metal-binding affinity, (2) provide a sensing mechanism against oxidative stress, and (3) aid in the regeneration of the protein. We believe that our findings open new directions of research in redox and metals biology.

# Materials and methods
## Materials
DEAE Sepharose CL-6B, Sephacryl S-100, Blue Sepharose 6 Fast Flow, Glutathione Sepharose 4 Fast Flow, and Benzamidine Sepharose 4 Fast Flow were obtained from GE Healthcare (Uppsala, Sweden). HPE-IAM was obtained from Molecular Biosciences (Boulder, CO, USA). Human recombinant Trx and insulin were purchased from Wako Pure Chemical Industries (Osaka, Japan). TR from rat liver (TrxB) was obtained from Sigma-Aldrich (St. Louis, MO, USA). pGEX-4T-1 vector was from GE Healthcare. Chemical synthesis of α-domain and β-domain of GIF/MT-3 proteins was carried out by HiPep Laboratories (Kyoto, Japan) using conventional solid-phase synthesis, followed by purification and characterization using LC–MS. The amino acid sequences of the proteins were identical to those shown in *Figure 3—figure supplement 1*. All other reagents and chemicals used were of the highest grade available.

## Protein expression and purification
Vector pEX-K4J2 containing human cDNA of MT-1A, MT-2, GIF/MT-3, or GIF/MT-3 mutants (all Cys-to-Ala, α-domain, β-domain, βCys-to-Ala, αCys-to-Ala, or all Ser-to-Ala) between the *Bam*HI and *Xho*I sites was obtained from Eurofins Genomics (Tokyo, Japan). The cDNAs were excised using *Bam*HI and *Xho*I and ligated into the corresponding sites of pGEX-4T-1, a glutathione *S*-transferase fusion expression vector. An overnight culture of *E. coli* BL21 (TOYOBO, Osaka, Japan) containing 1% vol/vol pGEX-4T1-1/cDNA vector in fresh Luria–Bertani medium was grown at 37°C for 2 hr, then ZnCl$_2$ (final concentration of 500 μM) and isopropyl-1-thio-β-D-galactopyranoside (final concentration of 100 μM) were added to induce the expression of the fusion protein. After 5 hr incubation at 37°C, cells were pelleted by centrifugation at 5000×$g$ for 10 min at 4°C and resuspended in 5% of the original volume of buffer A (20 mM Tris–HCl [pH 7.5], 150 mM NaCl), then lysed using mild sonication at 4°C. Triton X-100 was added to a final concentration of 1% vol/vol, and the suspension was mixed gently at approximately 20°C for 1 hr to facilitate protein solubilization. After centrifugation at 105,000×$g$ for 1 hr, the supernatant was applied at a flow rate of 2 mL/min to a Glutathione Sepharose column (5.5 cm × 1.5 cm i.d.) pre-equilibrated with buffer A. The column was washed with 100 mL of buffer A, then syringe-filled with 10 mL thrombin solution (400 U/mL in buffer A; Wako, Osaka, Japan) and incubated overnight at room temperature. After incubation, the target proteins and thrombin were eluted using buffer A. The eluate was filtered using a 3 kDa Amicon Ultra centrifugal filter unit (Millipore) following centrifugation with buffer C (20 mM Tris–HCl [pH 7.5], 500 mM NaCl) to concentrate the fractions and exchange the buffer. To remove thrombin, the concentrated sample was applied at a flow rate of 2 mL/min to a Benzamidine Sepharose 4 Fast Flow column (3.9 cm × 1.5 cm i.d.) pre-equilibrated with buffer C. The eluted sample was filtered eight times using a 3 kDa Amicon Ultra centrifugal filter unit with buffer B to exchange the buffer and remove small molecules, then stored at −80°C.

## Protein assay
The cytoplasmic protein concentration in mouse brain was determined using the bicinchoninic acid assay with bovine serum albumin as a standard. The concentrations of GIF/MT-3 protein were determined by measuring the absorbance of apo-GIF/MT-3 at 220 nm using an extinction coefficient of 53,000 M$^{-1}$cm$^{-1}$ (*Vasák, 1991*). To produce apo-GIF/MT-3, Zn$_7$GIF/MT-3-containing buffer was exchanged with 0.1 M HCl by ultrafiltration. Briefly, a protein solution (0.5 mL) was added to an Amicon Ultra centrifugal filter unit. After 30 min centrifugation at 14,000×$g$, the filtrate was discarded

and 0.1 M HCl (0.45 mL) was added to the retentate. This procedure was repeated two times, and the final retentate was exchanged with 20 mM Tris–HCl (pH 7.5) buffer and used for further studies.

## Protein isolation

Animal housing, husbandry, and euthanasia were conducted according to the guidelines of the Animal Care and Use Committee of the University of Tsukuba. Experimental protocols for mice were approved by the Animal Care and Use Committee of the University of Tsukuba (17-369). All surgery was performed under phenobarbital anesthesia, and every effort was made to minimize suffering. Approximately 50 C57BL/6 mice (10–20 weeks of age, ≈1:1 male/female), kindly provided by Prof. S Takahashi (University of Tsukuba), were anesthetized by intraperitoneal injection of 200 mg/kg phenobarbital. The number of mice was determined based on preliminary experiments that indicated the amount of tissue required for successful enzyme purification. Their brains were perfused with cold saline and then homogenized with four volumes of buffer A (20 mM Tris [pH 7.5], 150 mM NaCl). The homogenate was centrifuged at $9000 \times g$ for 10 min at 4°C, and the resulting supernatant was centrifuged at $105,000 \times g$ for 1 hr to obtain the cytosol. The cytosolic fraction was filtered and buffer-exchanged 27 times using a 3 kDa Amicon Ultra centrifugal filter unit and centrifugation at $5000 \times g$ for 30 min at 4°C with buffer B (20 mM Tris [pH 7.5]). To isolate SSBPs, the resulting high-molecular-weight fraction, containing 442 mg protein in 69 mL solution, was applied to a DEAE Sepharose CL-6B column (4.1 cm × 2.5 cm i.d.) pre-equilibrated with buffer B. The column was washed with buffer B at a flow rate of 1 mL/min, then SSBPs were eluted using 200 mL buffer B with a linear gradient of 0–0.4 M NaCl and 5 mL fractions were collected. The major SSBP-containing fractions (fractions 40–44 in *Figure 10A*) were filtered using a 3 kDa Amicon Ultra centrifugal filter unit and centrifugation at $14,000 \times g$ for 30 min at 4°C with buffer A seven times to concentrate the protein and exchange the buffer. The concentrated fraction (4.5 mL) was applied to a Sephacryl S-100 column (71 cm × 2.5 cm i.d.) previously equilibrated with buffer A and eluted with buffer A at a flow rate of 1 mL/min and 5 mL fractions were collected. The major SSBP-containing fractions with low total protein concentration (fractions 40–43 in *Figure 10B*) were combined and filtered, as described above, with buffer B. The concentrated fraction (1 mL) was applied to a Blue Sepharose column (8.4 cm × 2.5 cm i.d.) previously equilibrated with buffer B. The loaded column was washed with buffer B, then SSBPs were eluted at a flow rate of 1 mL/min using 200 mL buffer B with a linear gradient of 0–0.4 M NaCl and collected in 10 mL fractions. Eluate containing SSBP (fractions 3–5 in *Figure 10C*) was concentrated to a volume of 0.2 mL using Amicon Ultra centrifugal filter units. The resulting material was used for sulfane sulfur detection, SDS–PAGE with silver staining, and western blotting. All operations were performed at 4°C. Protein-bound sulfane sulfur was measured by determining bis-S-HPE-AM concentration after incubation of the protein with HPE-IAM. The incubation mixture (100 µL) consisted of 5 mM HPE-IAM and protein in buffer B. The reaction was performed at 37°C for 30 min, and the yield of the bis-S-HPE-AM adduct was determined using liquid chromatography–electrospray ionization–tandem mass spectrometry (LC–ESI–MS/MS), as described later in detail.

## Identification of proteins

To identify SSBP in the brain, the isolated protein (75 µg) was incubated with 20 mM dithiothreitol in 50 mM ammonium bicarbonate at 50°C for 1 hr, then incubated with 50 mM iodoacetamide in 50 mM ammonium bicarbonate at room temperature for 30 min, and finally digested with trypsin (1.2 µg) at 37°C overnight. The tryptic digests (2.5 µL) were loaded in direct injection mode onto a nanoAcquity ultra-performance LC system (Waters, Milford, MA, USA) equipped with a BEH130 nanoAcquity $C_{18}$ column (100 mm×75 µm, 1.7 µm i.d.) held at 35°C. Mobile phases A (0.1% vol/vol formic acid) and B (0.1% vol/vol formic acid in acetonitrile) were linearly mixed at a flow rate of 0.3 µL/min using a gradient system as follows: 3% B for 1 min; linear increase to 40% B over 65 min; linear increase to 95% B over 1 min; constant 95% B for 9 min; linear decrease to 3% B over 5 min. The total running time, including initial conditioning of the column, was 90 min. The eluted peptides were transferred to the nanoelectrospray source of a quadrupole TOF-MS instrument (Synapt High Definition Mass Spectrometry System, Waters) via a Teflon capillary union and pre-cut PicoTip (Waters). ESI was used with a capillary voltage of 3 kV and sampling cone voltage of 35 V. Low (6 eV) or elevated (step from 15 to 30 eV) collision energy was used to generate either intact peptide precursor ions (low energy) or peptide product ions (elevated energy). The source temperature was 100°C, and the detector was

operated in the positive-ion mode. Data were collected in the $m/z$=300–2000 range using an independent reference spray and the NanoLockSpray interference procedure in which Glu-1-fibrinopeptide B ($m/z$=785.8426) was infused via the NanoLockSpray ion source and sampled every 10 s for external mass calibration. Data were collected using MassLynx software (v4.1, Waters). ProteinLynx Global Server Browser (v2.3, Waters) was used to identify the protein based on its peptide mass fingerprints.

## Measurement of zinc concentration

Each sample was added to an acid-washed test tube containing nitric acid (0.3 mL) and $H_2O_2$ (0.1 mL) and digested at 130°C for 2 days in an aluminum dry bath block. The evaporated samples were dissolved in deionized distilled water, and zinc concentrations were measured using ICP–MS (ICPMS-2030, Shimadzu, Japan). A $ZnSO_4$ solution was used as a concentration standard.

## FT–ICR–MALDI–TOF/MS

Recombinant human $Zn_7$GIF/MT-3 was incubated with 0.1 N HCl at 37°C for 30 min and then exchanged with 20 mM Tris–HCl (pH 7.5) buffer at 37°C for 36 hr to prepare oxidized apo-GIF/MT-3. Low-molecular-weight molecules were then removed using a 3 kDa Amicon Ultra centrifugal filter unit. $Zn_7$GIF/MT-3 or apo-GIF/MT-3 (0.5 µL) in 20 mM Tris–HCl (pH 7.5) were mixed with a solution of α-cyano-4-hydroxycinnamic acid matrix (0.5 µL, 60% vol/vol acetonitrile, 0.2% vol/vol trifluoroacetic acid) and then dispensed into 384-well plates. The crystals obtained on the plate were analyzed using FT–ICR–MS (7T solariX, Bruker Daltonics, Billerica, MA, USA) equipped with a MALDI source operating in positive ion mode. Analytical conditions were as follows: $m/z$ range, 1000–10,000; number of scans averaged, 3; accumulation time, 2.00 s; polarity, positive.

## Raman spectroscopy

Recombinant $Zn_7$GIF/MT-3 was incubated with 0.1 N HCl for 30 min and then replaced with 20 mM Tris–HCl (pH 7.5) buffer for 36 hr at 37°C (apo-GIF/MT-3) or HPE-IAM (5 mM) in 20 mM Tris–HCl (pH 7.5) for 36 hr at 60°C. Then, low-molecular-weight molecules were removed using a 3 kDa Amicon Ultra centrifugal filter unit. The resulting protein was concentrated to ≈4 mg/mL using a 5 kDa centrifugal concentrator (Vivaspin; Sartorius, Göttingen, Germany). For drop-deposition Raman spectroscopy, 0.5 µL of each protein sample was first dried onto a hydrophobic quartz coverslip for up to 15 min under vacuum. Spectra were then collected from the 'coffee ring' of each drop, where proteins were found in the absence of bulk salt, using a Raman microscope system with a charge-coupled-device detector (InVia, Renishaw, New Mills, UK). Each sample was excited using a 785 nm diode laser focused through a Leica 50× (0.75 numerical aperture) short-working-distance air objective, with ≈100 mW power incident on each sample. The laser was focused onto the sample using an on-screen camera. WiRE software (v4.1, Renishaw) was used for spectral acquisition, data collection, and cosmic ray removal. The Raman system was calibrated against the 520 $cm^{-1}$ reference peak of silicon prior to each experiment. All spectra were processed using the IrootLab plugin (0.15.07.09-v) in MATLAB (The MathWorks, Inc, MA, USA). The background was carefully subtracted from the spectra using blank quartz spectra, then the background-corrected spectra were smoothed using a wavelet denoising function. Fluorescence was removed by fitting and subtracting a fifth-order polynomial, and the ends of each spectrum were anchored to the axis using a rubber-band-like function before intensity normalization.

## Detection of protein-bound sulfane sulfur

LC–ESI–MS/MS analysis with HPE-IAM was used to determine the levels of protein-bound sulfane sulfur. Some reagents such as $H_2O_2$, SNAP, TCEP, and KCN were filtered through a 3 kDa Amicon Ultra centrifugal filter unit prior to use. A high-molecular-weight cytosolic fraction from mouse brain or purified MT protein solution was incubated with HPE-IAM under the appropriate conditions to yield bis-S-HPE-AM adducts. Incubation with HPE-IAM at temperatures >60°C was not suitable because it yielded false positives for sulfane sulfur in protein-free negative controls. The resulting solutions were filtered through a 3 kDa Amicon Ultra centrifugal filter unit to obtain low-molecular-weight fractions containing bis-S-HPE-AM adducts. HPE-AM adducts were diluted fourfold with 0.1% vol/vol formic acid containing known amounts of isotope-labeled internal standard (bis-S34-HPE-AM) (*Akaike et al., 2017*) and the sulfane sulfur concentrations were determined using LC–ESI–MS/MS. A triple

quadrupole mass spectrometer (EVOQ Qube, Bruker) coupled to an ultra-high-performance LC system (Advance, Bruker) was used to perform LC–ESI–MS/MS. Sulfane sulfur-derived bis-S-HPE-AM was separated using a YMC-Triart $C_{18}$ column (50 mm× 2.0 mm i.d.) at 40°C. Mobile phases A (0.1% vol/vol formic acid) and B (0.1% vol/vol formic acid in methanol) were linearly mixed at a flow rate of 0.2 mL/min using a gradient system as follows: 3% B for 3 min; linear increase to 95% B over 12 min; constant 95% B for 1 min; linear decrease to 3% B. MS spectra were obtained using a heated ESI source with the following settings: spray voltage, 4000 V; cone temperature, 350°C; heated probe temperature, 250°C; cone gas flow, 25 psi; probe gas flow, 50 psi; nebulizer gas flow, 50 psi.

## Measurement of free SH/SSH content

Ellman's reagent (DTNB) was used to estimate the concentration of sulfhydryl groups in GIF/MT-3 by comparison with a standard curve of the sulfhydryl-containing compound GSH. Briefly, after removal of low-molecular-weight molecules, GIF/MT-3 protein (1 μM) was incubated with DTNB (500 μM) in 100 mM Tris–HCl (pH 8.0) and 1 mM EDTA at room temperature for 5 min, and the absorbance at 412 nm was measured.

## NADPH consumption

To study the kinetics of reduction of human apo-GIF/MT-3 by Trx, 200 μL reaction mixtures containing 100 mM KPi (pH 7.5), 100 μM NADPH, 1 μg human Trx (0.5 μM), 0.7 μg rat TrxB (50 nM), and the concentrations of oxidized apo-GIF/MT-3 indicated in *Figure 8A* were used. To study the reduction of human apo-MT-3 or human insulin by Trx, 200 μL reaction mixtures containing 100 mM KPi (pH 7.5), 100 μM NADPH, 1 μg human Trx (0.5 μM), 0.7 μg rat TrxB (50 nM), and the concentrations of oxidized apo-GIF/MT-3 or human insulin indicated in *Figure 8B* were used. To compare the reduction of human apo-GIF/MT-3 or $Zn_7GIF/MT-3$ by Trx, TRP14, or TRP32, 200 μL reaction mixtures containing 100 mM KPi (pH 7.5), 100 μM NADPH, 6 μg of human Trx, TRP14, or TRP32, 1 μg of rat TrxB, and 6 μg of oxidized apo-GIF/MT-3 or $Zn_7GIF/MT-3$ were used (*Figure 8C*). Reactions were performed at room temperature, and NADPH oxidation was monitored by measuring the absorbance of $NADP^+$ at 340 nm. To restore sulfane sulfur in apo-GIF/MT-3 after incubation with the Trx/TR system (*Figure 8D*), apo-GIF/MT-3 (5 μM) was incubated with Trx (6 μM), TrxB (72 nM), and NADPH (100 μM) at 25°C for 30 min in 100 mM KPi (pH 7.5) and then with 5 mM HPE-IAM. Sulfane sulfur content in apo-GIF/MT-3 was determined using LC–ESI–MS/MS after 3 kDa filtration, with the peak intensity obtained without apo-GIF/MT-3 being subtracted from that obtained with the complete mixture.

## Homology modeling of MT isoforms

Homology modeling of human GIF/MT-3 was performed using MOE software (2018.01; Chemical Computing Group ULC, 1010 Sherbrooke St. West, Suite #910, Montreal, QC, Canada, H3A 2R7, 2018). To construct a model of GIF/MT-3 (National Center for Biotechnology Information reference sequence: NP_005945.1), the crystal structure of rat MT-2 (PDB code: 4MT2) and the NMR structure of the α-domain of human GIF/MT-3 (PDB code: 2F5H) were used as templates. Following the alignment of the primary structures (*Figure 9—figure supplement 1A*), the sequence similarities of the templates were 67.7% and 100% compared with the β-domain and α-domain of GIF/MT-3, respectively. The metals in the templates were changed as desired or deleted to model apo-GIF/MT-3. To construct the GIF/MT-3 model, 100 independent models of the target protein were built using the segment-matching procedure in MOE. Refinement of the model with the lowest generalized Born/volume integral (GVBI) was achieved by energy minimization of outlier residues in Ramachandran plots generated within MOE. The final model of GIF/MT-3 exhibited a 3D structure similar to those of MT2 and the α-domain of GIF/MT-3 (*Figure 9—figure supplement 1B*). The same method was used to construct homology models of human MT-1A and MT-2, with rat MT-2 as the template for MT-1, and both rat MT-2 and the α-domain of human MT-2 (PDB code: 1MHU) as templates for MT2 (*Figure 9—figure supplement 2*).

## Generation of sulfane sulfur-bound 3D model of MT

All cysteine residues in the homology model of MT were replaced with cysteine persulfide or polysulfides by performing a residue scan using the Protein Design module of MOE, and the resultant changes in metal-binding affinities and complex stabilities were evaluated. The orientation of cysteine

persulfide was manually modified to increase its interaction with the metals. Supplementary file containing homology model of GIF/MT-3 with replacement of all cysteine residues by cysteine persulfide in PDB format (*.pdb) was linked to the article (*Supplementary file 1*).

### Protein thermostability and metal-binding affinity scoring

We assessed the influence of sulfane sulfur on protein unfolding free energy using the Protein Design module of MOE, which computed a stability scoring function, ΔΔGs, based on the GBVI and weighted surface area:

$$\Delta\Delta Gs = \alpha \left[ \Delta E_{vdw} + 0.5 \left( \Delta E_{coul} + \Delta E_{sol} \right) \right] + \beta \Delta E_{SS} + \gamma \Delta SA_{sc} + \varepsilon \Delta SA_{pol} - \Delta Gsu^{WT \rightarrow Mut}$$

where $\Delta E_{vdw}$ is the AMBER van der Waals interaction energy, $\Delta E_{coul}$ is the AMBER Coulomb interaction energy, $\Delta E_{sol}$ is the change in solvation energy calculated using the GBVI, and $E_{SS}$ is the change in energy due to the presence of a disulfide bond. $\Delta SA_{sc}$ and $\Delta SA_{pol}$ are the changes in the side-chain and polar surface areas, respectively. α is a scaling factor accounting for configurational entropy effects, and $\Delta Gsu^{WT \rightarrow Mut}$ is the change in stability of the unfolded states. The affinity score was also calculated using MOE software as the difference between the potential energy values of the protein, free zinc, and metal–protein complex.

### Quantum chemistry calculations

Since the whole structure of the GIF/MT-3 is quite large, we divided GIF/MT-3 into two domains (α-domain and β-domain) in the quantum chemistry calculations in order to assign the observed Raman spectra. We have independently constructed α-domain and β-domain models of apo-GIF/MT-3 with disulfide bonds between neighboring cysteines or tetrasulfide bonds. These models are referred to as apo-GIF/MT-3_S2 and apo-GIF/MT-3_S4, respectively, and are shown in *Figure 2B-F*. The initial structures were taken from the results of the homology modeling by MOE (see *Figure 9A*). H atoms were placed instead of Zn–S bonds in the models. To consider apo-GIF/MT-3 models, the nearest S atoms are supposed to form disulfide or tetrasulfide bonds. The Raman spectra were computed by frequency calculation. All quantum chemical calculations were carried out at B3LYP/6-31G(d) level by GAUSSIAN16 (Revision C.01, Gaussian, Inc, Wallingford CT, 2016). After obtaining the Raman spectra for the α- and β-domains, these spectra are summed to obtain the Raman spectra of the apo-GIF/MT-3 as illustrated in *Figure 1I*. The corresponding Zn-binding models were constructed and evaluated the Raman spectra in the same manner.

### Statistical analysis

The reported data represent the mean ± SD of three independent experiments, except for the MOE calculations. Technical replicates (triplicates) were used to assess measurement variability. The statistical significance of pair-wise differences was assessed using Student's t-test. p<0.05 was considered to indicate a statistically significant difference, and p<0.01 was considered highly significant.

## Acknowledgements

We thank Prof. T Sawa (Kumamoto University, Japan) for the kind donation of NAC derivatives and Prof. ESJ Arnér (Karolinska Institutet, Sweden) for the kind donation of TRP14 and TRP32. This work was supported by Grants-in-Aid for Scientific Research from the Ministry of Education, Culture, Sports, Science, and Technology of Japan (#18H05293 to YK; #20H04340, #21KK0207, #22H04799, and #22H05555 to YS).

## Additional information

### Funding

| Funder | Grant reference number | Author |
| --- | --- | --- |
| Japan Society for the Promotion of Science | #18H05293 | Yoshito Kumagai |

| Funder | Grant reference number | Author |
| --- | --- | --- |
| Japan Society for the Promotion of Science | #20H04340 | Yasuhiro Shinkai |
| Japan Society for the Promotion of Science | #21KK0207 | Yasuhiro Shinkai |
| Japan Society for the Promotion of Science | #22H04799 | Yasuhiro Shinkai |
| Japan Society for the Promotion of Science | #22H05555 | Yasuhiro Shinkai |

The funders had no role in study design, data collection and interpretation, or the decision to submit the work for publication.

## Author contributions

Yasuhiro Shinkai, Conceptualization, Data curation, Formal analysis, Funding acquisition, Validation, Investigation, Methodology, Writing – original draft, Writing – review and editing; Yunjie Ding, Data curation, Software, Formal analysis, Validation; Toru Matsui, George Devitt, Data curation, Software, Formal analysis; Masahiro Akiyama, Motohiro Nishida, Investigation; Tang-Long Shen, Resources, Investigation; Tomoaki Ida, Takaaki Akaike, Sumeet Mahajan, Investigation, Methodology; Jon M Fukuto, Investigation, Writing – review and editing; Yasuteru Shigeta, Software, Investigation, Methodology, Writing – review and editing; Yoshito Kumagai, Conceptualization, Supervision, Funding acquisition, Project administration, Writing – review and editing

## Author ORCIDs

Yasuhiro Shinkai ⓘ https://orcid.org/0000-0003-1772-5342
Motohiro Nishida ⓘ https://orcid.org/0000-0002-2587-5458
Takaaki Akaike ⓘ https://orcid.org/0000-0002-0623-1710
Yoshito Kumagai ⓘ https://orcid.org/0000-0003-4523-8234

## Ethics

Animal housing, husbandry, treatment, and euthanasia were conducted under the guidelines of the Animal Care and Use Committee of the University of Tsukuba. Experimental protocols for mice were approved by the Animal Care and Use Committee of the University of Tsukuba (17-369). All surgery was performed under phenobarbital anesthesia, and every effort was made to minimize suffering.

Reviewer #2 (Public review): https://doi.org/10.7554/eLife.92120.4.sa1
Reviewer #3 (Public review): https://doi.org/10.7554/eLife.92120.4.sa2
Author response https://doi.org/10.7554/eLife.92120.4.sa3

# Additional files

## Supplementary files

Supplementary file 1. Pdb file of $Zn_7S_{20}GIF/MT-3$ generated by homology modeling.
MDAR checklist

## Data availability

All data generated or analysed during this study are included in the manuscript and supporting files.

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
