## [Editor Report · eLife Assessment]

This **valuable** work provides **solid** evidence that a neuronal metallothionein, GIF/MT-3, incorporates metal-persulfide clusters. A variety of well-designed assays support the authors' hypothesis, revealing that sulfane sulfur is released from MT-3. However, the sufane sulfur content in the canonical induced MT-1 and MT-2 has not been demonstrated. Thus, the biological role of the persulfidated form is not yet clearly defined. There are caveats to the findings that limit the study, but the work will nevertheless prompt major follow-up work.

---

## [Referee Report · Reviewer #2 (Public review)]

Summary:

In this manuscript, the authors reveal that GIF/MT-3 regulates the zinc homeostasis depending on the cellular redox status. The manuscript technically sounds, and their data concretely suggest that the recombinant MTs, not only GIF/MT-3 but also canonical MTs such as MT-1 and MT-2, contain sulfane sulfur atoms for the Zn-binding. The scenario proposed by the authors seems to be reasonable to explain the Zn homeostasis by the cellular redox balance.

Strengths:

The data presented in the manuscript solidly reveal that recombinant GIF/MT-3 contains sulfane sulfur.

Weaknesses:

It remains unclear whether native MTs, in particular induced MTs in vivo contain sulfane sulfur or not.

Comments on revisions:

Although the authors have revealed the sulfane sulfur content in native MT-3, my question, namely, whether canonical MT-1 and MT-2 contained sulfane sulfur after the induction has been left.

The authors argue that the biological significance of sulfane sulfur in MTs lies in its ability to contribute to metal binding affinity, provide a sensing mechanism against oxidative stress, and aid in the regulation of the protein. Due to their biological roles, induced MT-1 and MT-2 could contain sulfane sulfur in their molecules. Thus, I expect the authors to evaluate or explain the sulfane sulfur content in induced MT-1 and MT-2.

---

## [Referee Report · Reviewer #3 (Public review)]

Summary:

The authors were trying to show that a novel neuronal metallothionein of poorly defined function, GIF/MT3, is actually heavily persulfidated in both the Zn-bound and apo (metal-free) forms of the molecule as purified from a heterologous (bacterial) or native host. Evidence in support of this conclusion is strong, with both spectroscopic and mass spectrometry evidence strongly consistent with this general conclusion. The authors would appear to have achieved their aims.

Strengths:

The analytical data in support of the author's primary conclusions are strong. The authors also provide some modeling evidence that supports the contention that MT3 (and other MTs) can readily accommodate a sulfane sulfur on each of the 20 cysteines in the Zn-bound structure, with little perturbation of the overall structure. This is not the case with Cys trisulfides, which suggests that the persulfide-metallated state is clearly positioned at lower energy relative to the immediately adjacent thiolate- or trisulfidated metal coordination complexes.

Weaknesses:

The biological significance of the findings is not entirely clear. On the one hand, the analytical data are solid (albeit using a protein derived from a bacterial over-expression experiment), and yes, it's true that sulfane S can protect Cys from overoxidation, but everything shown in the summary figure (Fig. 9D) can be done with Zn release from a thiol by ROS, and subsequent reduction by the Trx/TR system. In addition, it's long been known that Zn itself can protect Cys from oxidation. I view this as a minor shortcoming that will motivate follow-up studies.

Impact:

The impact will be high since the finding is potentially disruptive to the MT field for sure. The sulfane sulfur counting experiment (the HPE-IAM electrophile trapping experiment) may well be widely adopted by the field. Those in the metals field always knew that this was a possibility, and it will interesting to see the extent to which metal binding thiolates broadly incorporate sulfane sulfur into their first coordination shells.

Comments on revisions:

The revised manuscript is only slightly changed from the original, with the inclusion of a supplementary figure (Figure 3-figure supplement 2) and minor changes in the text. The authors did not choose to carry out the quantitative Zn binding experiment (which I really wanted to see), but given the complexities of the experiment, I'll let it go.

---

## [Author Response]

The following is the authors’ response to the previous reviews

**Reviewer #2 (Public Review):**
Comments on revisions:Although the authors have revealed the sulfane sulfur content in native MT-3, my question, namely, whether canonical MT-1 and MT-2 contained sulfane sulfur after the induction has been left.The authors argue that the biological significance of sulfane sulfur in MTs lies in its ability to contribute to metal binding affinity, provide a sensing mechanism against oxidative stress, and aid in the regulation of the protein. Due to their biological roles, induced MT-1 and MT-2 could contain sulfane sulfur in their molecules. Thus, I expect the authors to evaluate or explain the sulfane sulfur content in induced MT-1 and MT-2.

Thank you for your valuable comments. In this study, we were not able to examine the role of sulfane sulfur in the induced forms of MT-1 and MT-2. However, this topic is undoubtedly important and intriguing; therefore, we will continue to explore it in future studies.

**Reviewer #3 (Public Review):**
Comments on revisions:The revised manuscript is only slightly changed from the original, with the inclusion of a supplementary figure (Fig. S2) and minor changes in the text. The authors did not choose to carry out the quantitative Zn binding experiment (which I really wanted to see), but given the complexities of the experiment, I'll let it go.Fig. 9: the authors imply in the mechanistic "redox-switch" figure that Trx/TR can not reduce persulfide linkages. A number of groups have shown this to be the case. I recommend modifying the figure legend or text to make this clear to the reader.

Thank you for your understanding. Regarding the "redox-switch" figure, although some groups have demonstrated the ability of Trx to reduce persulfide moieties, as you pointed out, we have addressed this discrepancy in the Discussion section as follows (lines 357-361): “In contrast, Trx has been proposed to reduce the persulfide moiety of PTP1B (37) and albumin (38, 39). A possible explanation for this discrepancy is that apo-GIF/MT-3-persulfide is rapidly changed into a different conformation that is topologically resistant to Trx reduction. In other words, Trx may exhibit substrate specificity.” Additionally, we have inserted the following sentence just before the above discussion to further clarify this point:“This suggests that the persulfide moiety in GIF/MT-3 appears to be relatively stable against Trx reduction.”